# Rao-Blackwell Gradient Estimators for Equivariant Denoising Diffusion

**Vinh Tong**[1,2]*, **Trung-Dung Hoang**[4]*, **Anji Liu**[5], **Guy Van den Broeck**[3], **Mathias Niepert**[1,2]
[1]University of Stuttgart, [2]IMPRS-IS, [3]UCLA, [4]University of Bern, [5]National University of Singapore
vinh.tong@ki.uni-stuttgart.de

## Abstract

In domains such as molecular and protein generation, physical systems exhibit inherent symmetries that are critical to model. Two main strategies have emerged for learning invariant distributions: designing equivariant network architectures and using data augmentation to approximate equivariance. While equivariant architectures preserve symmetry by design, they often involve greater complexity and pose optimization challenges. Data augmentation, on the other hand, offers flexibility but may fall short in fully capturing symmetries. Our framework enhances both approaches by reducing training variance and providing a provably lower-variance gradient estimator. We achieve this by interpreting data augmentation as a Monte Carlo estimator of the training gradient and applying Rao–Blackwellization. This leads to more stable optimization, faster convergence, and reduced variance, all while requiring only a single forward and backward pass per sample. We also present a practical implementation of this estimator—incorporating the loss and sampling procedure—through a method we call *Orbit Diffusion*. Theoretically, we guarantee that our loss admits equivariant minimizers. Empirically, Orbit Diffusion achieves state-of-the-art results on GEOM-QM9 for molecular conformation generation, improves crystal structure prediction, and advances text-guided crystal generation on the Perov-5 and MP-20 benchmarks. Additionally, it enhances protein designability in protein structure generation. Code is available at https://github.com/vinhsuhi/Orbit-Diffusion.git.

## 1 Introduction

Diffusion models have emerged as powerful methods for modeling complex distributions (Ho et al., 2020; Song et al., 2021a,b; Karras et al., 2022), with applications in domains such as molecular and protein generation (Zhang et al., 2023; Vignac et al., 2023; Anand & Achim, 2022). Many physical systems, such as molecules or crystals, exhibit inherent symmetries. For example, a molecule's physical properties remain unchanged under rotations in 3D space (Hoogeboom et al., 2022; Jing et al., 2022). Modeling such data requires learning distributions that are invariant under the action of a group $G$. This setting is naturally captured by the notion of $G$-*invariant* distribution $q(x_0)$ that are invariant under transformations from $G$ (Chen et al., 2024; Köhler et al., 2020).

Two main strategies have emerged for learning $G$-*invariant* distributions: (1) designing equivariant network architectures, and (2) using data augmentation to approximate equivariance. Equivariant architectures, such as equivariant denoisers, ensure symmetry by construction (Hoogeboom et al., 2022; Klein et al., 2024; Igashov et al., 2024), but are often less efficient due to increased architectural complexity and can pose optimization challenges (Brehmer et al., 2024; Abbe & Boix-Adserà, 2022). In contrast, data augmentation is a flexible and widely used alternative that approximates equivariant training by sampling transformed versions of the input. While this approach scales easily and has

---

*Equal contribution.

become increasingly popular in the community, especially for large-scale models (Abramson et al., 2024b; Geffner et al., 2025), its effectiveness in capturing symmetries may vary depending on the group and application domain. In some settings, such as molecular dynamics and structural biology, explicit equivariance remains beneficial (Anand & Achim, 2022; Batatia et al., 2022; Zaverkin et al., 2024). In this work, we propose a framework that provably improves *both* approaches: we introduce a novel form of implicit data augmentation by computing the denoising target as a weighted average over group orbits, which reduces variance and improves the training of equivariant denoisers.

We revisit data augmentation from a principled perspective and interpret it as a Monte Carlo estimator of the gradient of a symmetrized loss. This loss is defined over a fully symmetrized dataset, which yields an empirical distribution invariant under $G$ (Chen et al., 2024). While traditional data augmentation uses one or few samples from this augmented dataset, we instead apply Rao–Blackwellization to derive a new estimator with provably lower variance. Specifically, we decompose the gradient as an outer expectation over data and an inner conditional expectation over group actions. Replacing the noisy sample-based target with its conditional expectation yields a lower-variance gradient estimator, while preserving equivariance.

To translate our theoretical insights into a practical method, we develop an efficient implementation of the proposed gradient estimator. This approach integrates the symmetrized loss and variance reduction into a modified training objective, without increasing the computational cost. Our implementation requires only a single forward and backward pass per sample, and it is compatible with both equivariant and non-equivariant architectures. We refer to this practical method as *Orbit Diffusion*. By implicitly applying Rao–Blackwellization through a tailored loss formulation and sampling scheme, Orbit Diffusion enables stable optimization and improved generalization across a wide range of symmetry groups and tasks.

We provide theoretical guarantees that our symmetrized loss admits equivariant minimizers and that our gradient estimator has strictly smaller variance compared to existing methods. Empirically, we demonstrate strong performance across multiple domains. Our method achieves state-of-the-art results on GEOM-QM9 for molecular conformation generation, enhances crystal structure prediction, and improves text-guided crystal generation on the Perov-5 and MP-20 benchmarks. Moreover, our approach is compatible with non-equivariant denoisers; in particular, it improves the designability of protein structures generated by PROTEINA (Geffner et al., 2025).

## 2 Background

**Groups.** A *group* is a mathematical structure comprising a set $G$ and a binary operation $m : G \times G \to G$ that combines two elements of $G$. A *group action* $g \in G$ defines how the group $G$ acts on a set $\Omega$, such as a set of geometric objects.[2] We restrict our attention to *locally compact isometry groups*. Locally compact isometry groups encompass a broad class of transformation groups that preserve distances and possess a well-behaved topological structure. Examples include the permutation group $\mathsf{S}_n$, the orthogonal group $\mathsf{O}(d)$, and the special orthogonal group $\mathsf{SO}(d)$.

**Invariance and Equivariance.** A function $f : \Omega \to \mathbb{R}$ is said to be *G-invariant* if for all $g \in G$ and $x \in \Omega$, it satisfies $f(g \circ x) = f(x)$. This means the function value does not change under the action of any group element. A function $f : \Omega \to \Omega$ is said to be *G-equivariant* if for all $x \in \Omega$, $f$ commutes with any group action $g \in G$: $f(g \circ x) = g \circ f(x)$.

**Invariant and Equivariant Distributions.** A probability distribution $p(x)$ defined on a set $\Omega$ is said to be *G-invariant* under the action of a group $G$ if the probability of any measurable subset $A \subseteq \Omega$ remains unchanged under the transformation induced by any group action $g \in G : p(g \circ x \in A) = p(x \in A)$. A conditional distribution $p(y \mid x)$, where $x \in \Omega$ and $y \in \Omega$, is said to be *G-equivariant* if for all $g \in G$, the following condition holds: $p(g \circ y \mid g \circ x) = p(y \mid x)$..

**Group Symmetrization.** Let $S_G$ be the symmetrization operator under group $G$, transforming any distribution $p(x)$ into a *G-invariant*-invariant distribution, denoted as the *G-symmetrized* distribution:

$$S_G[p](x) := \int_G p(g \circ x) \, \mathrm{d}\mu_G(g), \quad \text{where } \mu_G \text{ is the Haar measure on } G. \tag{1}$$

---

[2]When the group acts on a vector space $V$, we do not distinguish between the abstract group element $g \in G$ and its linear representation $\rho(g) : G \to \mathrm{GL}(V)$. For simplicity, we write $g \circ x$ to denote the action $\rho(g)(x)$.

**Diffusion Models and Equivariant Diffusion Models.** Diffusion models (Ho et al., 2020) are a class of generative models that construct complex data distributions by iteratively transforming simple noise distributions through a learned denoising process. Formally, given data $x_0 \sim q(x_0)$, the forward process generates a sequence $x_t$ over time $t \in [0, T]$ using a stochastic differential equation (SDE) (Song et al., 2021b) or a discrete Markov chain (Ho et al., 2020), such as:

$$x_t = \alpha_t x_0 + \sigma_t \epsilon, \quad \epsilon \sim \mathcal{N}(0, I),$$

where $\alpha_t$ and $\sigma_t$ are a time-dependent scaling factor and a noise factor that determines the level of noise added at each step, respectively. The noise should be increasingly added to the sample so that at time $t = T$, $q(x_T) \approx \mathcal{N}(0, I)$.

The reverse process, parameterized by a neural network $\phi_\theta(x_t, t)$, approximates a clean sample $x_0$ given its noisy version $x_t$. The training objective typically involves minimizing a reweighted form of the denoising loss (Ho et al., 2020; Song et al., 2021b; Karras et al., 2022):

$$\mathcal{L} = \mathbb{E}_{t \sim \mathcal{U}(0,T),(x_0,x_t) \sim q(x_0,x_t)} \left[ \omega(t) \| \phi_\theta(x_t, t) - x_0 \|^2 \right], \tag{2}$$

where $\omega(t)$ is a time-dependent loss weight. For notational simplicity, we omit this term throughout the remainder of the paper. To sample from the diffusion model, we begin with a noise vector $x_T \sim \mathcal{N}(0, I)$ and iteratively apply the learned reverse process to transform it into a data sample $x_0$ using the trained model $\phi_\theta$. The reverse process can involve solving an ODE or SDE numerically (Song et al., 2021b; Karras et al., 2022; Lu et al., 2022; Tong et al., 2025).

Equivariant diffusion models extend standard diffusion by enforcing equivariance of the neural network denoiser. Specifically, the denoiser $\phi_\theta$ is said to be *G-equivariant* if it satisfies $\phi_\theta(g \circ x_t, t) = g \circ \phi_\theta(x_t, t)$ for all $g \in G$.

## 3 Method

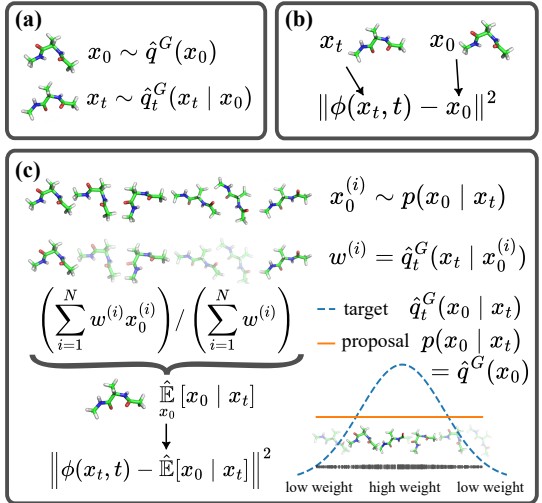

Figure 1: Gradient estimation strategies for training (approximately) equivariant diffusion models: (a) Sampling from the symmetrized joint distribution to obtain $x_0$ and $x_t$. (b) The standard data augmentation approach, which directly uses these samples for training. (c) The proposed method, leveraging self-normalizing importance sampling (SNIS) to estimate the inner conditional expectation. Both (b) and (c) require a single neural function evaluation per gradient step, but (c) has lower variance than (b). The pseudo-code for the Rao-Blackwell estimator with SNIS is shown on the right.

Let $G$ be a symmetry group, such as the group of Euclidean rotations. Our goal is to learn a $G$-invariant data-generating distribution $q(x_0)$. However, the observed distribution $\hat{q}$ (from which we obtain training samples) is generally not $G$-invariant due to dataset biases. For example, in molecular datasets, each molecule may be stored in a canonical but arbitrary orientation, even though physically

all rotated versions are equally probable under $q$. As a result, training directly on $\hat{q}$ would lead to a model that may not respect the underlying symmetry.

This issue is widely recognized in the literature, and two standard solutions are:

1. **Equivariant model:** Train with the diffusion loss in Equation (2) using a $G$-equivariant network $\phi(x_t, t)$.

2. **Data augmentation:** Train a non-equivariant $\phi$ with Equation (2), augmenting data with random group actions to encourage approximate $G$-invariance.

The second approach has been widely adopted—several high-profile non-equivariant models achieve strong results through augmentation (Abramson et al., 2024b; Geffner et al., 2025). However, empirical evidence shows that augmentation offers *no benefit* for already equivariant models, a result we formally prove in Appendix B.2.

### 3.1 From Symmetrized Loss to High-Variance Gradient Estimators

To unify these approaches, consider the *symmetrized data distribution* from Equation (1), and the forward noising kernel $\hat{q}_t^G(x_t \mid x_0)$ (e.g., a Gaussian in denoising diffusion) with marginal $\hat{q}_t^G(x_t)$. The *symmetrized diffusion loss* at time $t$ is

$$\mathcal{L}_t^G(\phi) = \mathbb{E}_{x_0 \sim \hat{q}^G} \, \mathbb{E}_{x_t \sim \hat{q}_t^G(\cdot \mid x_0)} \big[ \| \phi(x_t, t) - x_0 \|^2 \big]. \tag{3}$$

Both the equivariant model and the data augmentation approach can be viewed as implicitly minimizing $\mathcal{L}_t^G(\phi)$ (Chen et al., 2024). The *true* gradient of this loss is

$$\nabla_\phi \mathcal{L}_t^G = \mathbb{E}_{x_t \sim \hat{q}_t^G} \, \mathbb{E}_{x_0 \sim \hat{q}_t^G(\cdot \mid x_t)} \big[ 2 \big( \phi(x_t, t) - x_0 \big) \big], \tag{4}$$

where the expectations are taken over the full symmetrized joint distribution.

In practice, we do not have access to the exact expectations in Equation (4). Instead, we construct a Monte Carlo *gradient estimator* by sampling $x_0 \sim \hat{q}$ (or $\hat{q}^G$ in the augmented/equivariant case), then sampling $x_t \sim \hat{q}_t^G(\cdot \mid x_0)$, and using the single-sample estimate $\widehat{\nabla_\phi \mathcal{L}_t^G} = 2 \big( \phi(x_t, t) - x_0 \big)$. This estimator is *unbiased*, but, as in other diffusion training setups, it can exhibit *high variance* (Kingma et al., 2021; Xu et al., 2023; Nichol & Dhariwal, 2021). Increasing the batch size reduces variance but requires more forward passes of the neural network $\phi$, raising computational cost.

In this work, we introduce a class of Rao–Blackwellized gradient estimators that *provably reduce variance* while remaining unbiased, and can be applied to both the equivariant and augmentation-based training strategies.

### 3.2 Rao–Blackwellized Gradient Estimator

Our *key observation* is that $\phi(x_t, t)$ does not depend on $x_0$. This allows us to move it outside the inner expectation in Equation (4), yielding

$$\nabla_\phi \mathcal{L}_t^G = \mathbb{E}_{x_t \sim \hat{q}_t^G} \Big[ 2 \big( \phi(x_t, t) - \underbrace{\mathbb{E}_{x_0 \sim \hat{q}_t^G(x_0 \mid x_t)} [x_0]}_{\mathbb{E}[x_0 \mid x_t]} \big) \Big]. \tag{5}$$

Replacing $x_0$ with its conditional mean $\mathbb{E}[x_0 \mid x_t]$ yields a *Rao–Blackwellized (RB) gradient estimator*, which remains unbiased and has variance no greater than the original—strictly less unless $x_0 \mid x_t$ is deterministic, a situation that rarely occurs in generative modeling where $x_0$ is typically stochastic given $x_t$. This improvement requires no additional neural network evaluations, only a more accurate target. To avoid custom backward passes, we can minimize:

$$\mathcal{L}_t^{\mathrm{RB}}(\phi) = \mathbb{E}_{x_t \sim \hat{q}_t^G} \Big[ \| \phi(x_t, t) - \mathbb{E}[x_0 \mid x_t] \|^2 \Big] \tag{6}$$

> **Variance reduction guarantee**
>
> **Theorem 1.** *Let $\widehat{\nabla}_\phi$ be the Monte Carlo gradient from Equation* (4) *and $\widehat{\nabla}_\phi^{(RB)}$ be from Equation* (5). *If $\mathbb{E}[x_0 \mid x_t]$ can be computed exactly, then*
>
> $$\mathrm{Var}\big(\widehat{\nabla}_\phi^{(RB)}\big) \leq \mathrm{Var}\big(\widehat{\nabla}_\phi\big),$$
>
> *with strict inequality unless $x_0 \mid x_t$ is a Dirac delta.*

The challenge now is estimating the loss target $\mathbb{E}[x_0 \mid x_t]$ accurately and efficiently. We address this next using self-normalized importance sampling (SNIS).

## 3.3 Estimating the Conditional Expectation

A central challenge in computing the gradient estimator is evaluating the conditional expectation $\mathbb{E}_{x_0 \sim \hat{q}_t^G(x_0 \mid x_t)}[x_0]$, which is generally intractable:

$$\mathbb{E}_{x_0 \sim \hat{q}_t^G(x_0 \mid x_t)}[x_0] = \int_\Omega x_0 \, \hat{q}_t^G(x_0 \mid x_t) \, \mathrm{d}x_0. \tag{7}$$

This expectation can be approximated by drawing independent samples $x_0^{(1)}, \ldots, x_0^{(N)} \sim \hat{q}_t^G(x_0 \mid x_t)$ and computing the sample mean. Unfortunately, we cannot sample efficiently and directly from $\hat{q}_t^G(x_0 \mid x_t)$. Using Bayes' rule:

$$\hat{q}_t^G(x_0 \mid x_t) \propto \hat{q}_t^G(x_t \mid x_0)\hat{q}^G(x_0), \tag{8}$$

where $\hat{q}_t^G(x_t \mid x_0)$ is available in closed form, but $\hat{q}^G(x_0)$ is intractable due to integration over the group orbit of $x_0$.

To address this, we use self-normalized importance sampling (SNIS) with a proposal distribution $p(x_0 \mid x_t)$ that shares the same intractable orbit-integral structure as $\hat{q}^G(x_0)$, allowing cancellation of the problematic terms in the importance weights. The conditional expectation is approximated as:

$$\mathbb{E}_{x_0 \sim \hat{q}_t^G(x_0 \mid x_t)}[x_0] \approx \frac{\sum_{i=1}^N x_0^{(i)} \cdot w^{(i)}}{\sum_{i=1}^N w^{(i)}}, \quad w^{(i)} = \hat{q}_t^G(x_t \mid x_0^{(i)}) \cdot \frac{\hat{q}^G(x_0^{(i)})}{p(x_0^{(i)} \mid x_t)}. \tag{9}$$

The design of the proposal $p(x_0 \mid x_t)$ aims to ensure that the quotient $\hat{q}^G(x_0)/p(x_0 \mid x_t)$ becomes tractable. This can be achieved by first sampling from the original dataset $D$ using some user-defined $\bar{p}(x_0 \mid x_t)$, where $\bar{p}$ is designed to have non-zero probability for all elements in the dataset. Once a sample $x_0 \in D$ is drawn, we sample a group element $g$ from the group $G$ uniformly at random, and apply this group action to the sample. This results in a new sample $x_0^{(i)} = g \circ x_0$. This proposal distribution inherits the same orbit-integral structure as $\hat{q}^G(x_0)$, causing the intractable terms in the ratio $\hat{q}^G(x_0^{(i)})/p(x_0^{(i)} \mid x_t)$ to cancel. Specifically, with $\delta$ the Dirac delta function:

$$\frac{\hat{q}^G(x_0^{(i)})}{p(x_0^{(i)} \mid x_t)} = \frac{\hat{q}^G(g \circ x_0)}{p(g \circ x_0 \mid x_t)} = \frac{\hat{q}(x_0) \int_G \delta(g \circ x_0 - g' \circ x_0) \, \mathrm{d}\mu_G(g')}{\bar{p}(x_0 \mid x_t) \int_G \delta(g \circ x_0 - g' \circ x_0) \, \mathrm{d}\mu_G(g')} = \frac{\hat{q}(x_0)}{\bar{p}(x_0 \mid x_t)}. \tag{10}$$

Beside, since $\hat{q}(x_0) = 1/|D|$ for any $x_0 \in D$, this term can be omitted from the importance weight. Thus, the final importance weight simplifies to $w^{(i)} = \hat{q}_t^G(x_t \mid x_0^{(i)})/\bar{p}(x_0 \mid x_t)$. Importantly, all components of the importance weights are tractable: $\hat{q}_t^G(x_t \mid x_0^{(i)})$ is the forward diffusion process; $\hat{q}(x_0)$ corresponds to the empirical data distribution; and $\bar{p}(x_0 \mid x_t)$ is user-defined and tractable. Moreover, SNIS estimators based on these importance weights are always consistent.

An important instance results from setting $\bar{p}(x_0 \mid x_t) = \hat{q}(x_0)$ where the proposal recovers the exact symmetrized distribution: $p(x_0 \mid x_t) = \hat{q}^G(x_0)$ and the importance weight simplifies to $w^{(i)} = \hat{q}_t^G(x_t \mid x_0^{(i)})$.

## 3.4 Practical Implementation—Orbit Diffusion (`OrbDiff`)

We present *Orbit Diffusion* (`OrbDiff`) as a practical variant of our estimator. Although using $\hat{q}^G(x_0)$ as the proposal is theoretically valid, it is inefficient and cumbersome in practice. For small $t$, the conditional $\hat{q}_t^G(x_t \mid x_0)$ is sharply concentrated around the $x_0$ that generated $x_t$, so uniformly sampled $x_0$ rarely yield useful gradients. Furthermore, sampling from the full support generally requires drawing points outside the current minibatch, adding non-trivial implementation complexity.

To improve efficiency, we fix $x_0$ to the example that produced $x_t$ and sample candidates only from its *orbit* $\mathcal{O}_{x_0} = \{g \circ x_0 \mid g \in G\}$. This biases the proposal toward points with high likelihood under $\hat{q}_t^G(x_t \mid x_0)$—for instance, small rotations in $\mathsf{SO}(3)$-equivariant settings or local permutations in discrete symmetry groups. Since such points dominate the conditional distribution at small noise levels, orbit sampling greatly improves sample efficiency by prioritizing candidates with non-trivial importance weights. At high noise levels, contributions from outside the orbit may increase, but their weights are typically small, and expanding the proposal has shown little benefit.

Formally, `OrbDiff` replaces the intractable conditional expectation $\mathbb{E}[x_0 \mid x_t]$ in Equation (6) with the *orbit-weighted target*

$$\phi^*(x_0, x_t, t) = \frac{1}{Z(x_t, x_0)} \int_G (g \circ x_0)\, \hat{q}_t^G(x_t \mid g \circ x_0)\, \mathrm{d}\mu_G(g), \tag{11}$$

where $Z(x_t, x_0) = \int_G \hat{q}_t^G(x_t \mid g \circ x_0)\, \mathrm{d}\mu_G(g)$ is the normalization constant. This yields the `OrbDiff` loss:

$$\mathcal{L}_t^{\texttt{OrbDiff}}(\phi) = \mathbb{E}_{x_0 \sim \hat{q}^G(x_0)} \mathbb{E}_{x_t \sim \hat{q}_t^G(\cdot \mid x_0)} \left[ \|\phi(x_t, t) - \phi^*(x_0, x_t, t)\|^2 \right]. \tag{12}$$

Figure 1 provides an illustration and pseudo-code of the practical implementation. When $\phi$ is $G$-equivariant (see Appendix B.2 and Chen et al. (2024)), the loss in Equation (12) and its gradient are unchanged if $x_0$ is drawn from the empirical distribution $\hat{q}(x_0)$ rather than $\hat{q}^G(x_0)$.

Even though $\phi^*(x_0, x_t, t) \neq \mathbb{E}[x_0 \mid x_t]$, for an equivariant forward process the gradient of Equation (12) matches that of Equation (5), ensuring that `OrbDiff` yields an unbiased gradient estimate. The orbit-weighted target is also equivariant, providing a training signal aligned with the model's inductive bias. We formally prove both properties in Appendix B.4.

---

**Unbiased gradient and equivariance of `OrbDiff`**

**Theorem 2.** *Let $G$ be a locally compact isometry group acting on data space $\Omega$, and suppose the forward kernels $\hat{q}_t^G(x_t \mid x_0)$ are $G$-invariant: $\hat{q}_t^G(g \circ x_t \mid g \circ x_0) = \hat{q}_t^G(x_t \mid x_0)$ for all $g \in G$. Then:*

1. *The `OrbDiff` target $\phi^*(x_0, x_t, t)$ satisfies: $\phi^*(x_0, h \circ x_t, t) = h \circ \phi^*(x_0, x_t, t)$ for all $h \in G$.*
2. *The gradient of the `OrbDiff` loss (12) equals that of the ideal loss (5), i.e., `OrbDiff` provides an unbiased gradient estimator.*

---

We also explore non-uniform sampling of group elements $g$ for approximating the conditional expectation. For small noise, we sample near the identity action, expanding the neighborhood as noise increases. These distributions, $\nu_t(g)$, depend on the noise schedule. While not all groups support closed-form expressions for the density of individual group elements, they exist for the translation group (sampled from a Gaussian) or $\mathsf{SO}(3)$ (sampled from the von Mises-Fisher distribution).

Next, we divide by $\nu_t(g)$ to account for the group sampling distribution, resulting in the importance weights $w^{(i)} = \hat{q}_t^G(x_t \mid x_0^{(i)})/\nu_t(g)$. We also ensure that the identity group element is included in the sampled set. This strategy is effective in practice and provides computational advantages.

## 4 Experimental Results

Our experiments evaluate the generality and robustness of Orbit Diffusion across diverse generative tasks. We begin with a controlled synthetic setup using a standard diffusion model (Section 4.1) and

consider various isometry groups, including reflections, rotations, translations, and graph automorphisms. We then extend our method to Flow Matching (Section 4.2) and to diffusion models with non-standard forward processes (Section 4.3). Finally, we apply Orbit Diffusion to a non-equivariant denoiser, demonstrating its effectiveness without architectural symmetry (Section 4.4).

## 4.1 Experiments on Synthetic Data

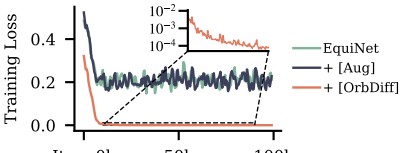

Figure 2: Learning curves.

| Variant | RMSD ($\times 10^{-5}$) ($\downarrow$) | W2 ($\times 10^{-3}$) ($\downarrow$) |
|---|---|---|
| EquiNet | $9.05 \pm 9.47$ | $1.150 \pm 1.581$ |
| + [Aug] | $9.50 \pm 9.63$ | $0.853 \pm 1.156$ |
| + [OrbDiff] | $\mathbf{0.37 \pm 0.09}$ | $\mathbf{0.004 \pm 0.001}$ |

Table 1: Synthetic experiment results: RMSD to the closest target in $\{-1, 1\}$ and W2 distance to ground-truth distribution.

We construct a dataset with a single 1D sample $x_0 = 1$, where the equivariant group is reflection: $g \circ x = \pm x$. We train a denoiser (EquiNet) of the form $D_\theta(x_t, t) = f_\theta(x_t, t) - f_\theta(-x_t, t)$, which is equivariant by design, with $f_\theta$ being a simple 3-layer MLP.

We evaluate three training variants of EquiNet: default, trained without Orbit Diffusion or data augmentation; + [Aug], with data augmentation only; and + [OrbDiff], with Orbit Diffusion only. Each model is trained for 100k iterations. After training, we generate 100k samples and compute the root mean square deviation (RMSD) from the closest target in $\{-1, 1\}$, and the Wasserstein-2 (W2) distance to the target distribution. We report mean and standard deviation for each model in Table 1.

Figure 2 shows the training loss curves for all three EquiNet variants. EquiNet and its augmented version exhibit similar fluctuations and magnitudes, indicating that data augmentation does not reduce variance. This is consistent with our theoretical result (Appendix B.2): for an equivariant denoiser, augmented and non-augmented losses are equivalent. In contrast, the OrbDiff variant shows a smoother and lower loss curve, confirming that Rao-Blackwellization reduces gradient variance and stabilizes training. As shown in Table 1, OrbDiff achieves roughly 25× lower RMSD and 200× lower W2 distance, outperforming both EquiNet variants.

## 4.2 Molecular Conformer Generation

**Molecular Conformer Generation.** Molecular Conformer Generation aims to generate plausible 3D structures from 2D molecular graphs, crucial for drug discovery and property prediction due to the role of 3D geometry (Liu et al., 2023; Axelrod & Gómez-Bombarelli, 2020).

We evaluate on the GEOM-QM9 dataset (Axelrod & Gomez-Bombarelli, 2022), respecting two key symmetries: invariance under global 3D rotations and equivariance to graph automorphisms, which permute atom indices without altering molecular identity. We compare against strong baselines, including GEOMOL (Ganea et al., 2021), Torsional Diffusion (Jing et al., 2022), MCF (Wang et al., 2024b), and ETFLOW (Hassan et al., 2024). Our method, Orbit Diffusion, is integrated into ETFLOW, a strong equivariant flow matching model that employs a harmonic prior for bonded atom proximity. We finetune ETFLow with OrbDiff using their public checkpoint.

During training, we apply symmetry-aware sampling by uniformly sampling 50 automorphisms and 200 SO(3) rotations per molecule, including the identity. These are applied to both 2D graphs and 3D conformers. All other settings follow ETFLOW; see Appendix C.1 for details.

We benchmark against three versions of ETFLOW: the results reported in the original paper, their released checkpoint, and our own reproduced results using the provided code and configuration [3]. Despite extensive effort, we were unable to match their reported performance, so we report all results under the same evaluation protocol.

---

[3]https://github.com/shenoynikhil/ETFlow

Table 2: Molecular conformer generation performance on GEOM-QM9. [*] Reported in the original paper. [†] Obtained using the published checkpoint. [‡] We train the public implementation from scratch.

| Models | Recall | | | | | | Precision | | | | | |
|---|---|---|---|---|---|---|---|---|---|---|---|---|
| | Cov@0.1 (↑) | | Cov@0.5 (↑) | | AMR (↓) | | Cov@0.1 (↑) | | Cov@0.5 (↑) | | AMR (↓) | |
| | Mean | Median | Mean | Median | Mean | Median | Mean | Median | Mean | Median | Mean | Median |
| GEOMOL[†] | 28.4 | 0.0 | 91.1 | 100.0 | 0.224 | 0.194 | 20.7 | 0.0 | 85.8 | 100.0 | 0.271 | 0.243 |
| Torsional Diff.[†] | 37.7 | 25.0 | 88.4 | 100.0 | 0.178 | 0.147 | 27.6 | 12.5 | 84.5 | 100.0 | 0.221 | 0.195 |
| MCF[†] | 81.9 | 100.0 | 94.9 | 100.0 | 0.103 | 0.049 | 78.6 | 93.8 | 93.9 | 100.0 | 0.113 | 0.055 |
| ETFLOW[*] | - | - | 96.5 | 100.0 | 0.073 | 0.047 | - | - | 94.1 | | 0.098 | 0.039 |
| ETFLOW[†] | 79.5 | 100.0 | 93.8 | 100.0 | 0.096 | 0.037 | 74.4 | 83.3 | 88.7 | 100.0 | 0.142 | 0.066 |
| ETFLOW[‡] | 81.4 | 100.0 | 94.4 | 100.0 | 0.092 | 0.039 | 74.6 | 85.5 | 89.1 | 100.0 | 0.145 | 0.064 |
| +[OrbDiff] | 85.4 | 100.0 | 96.3 | 100.0 | 0.074 | 0.027 | 80.2 | 93.9 | 91.9 | 100.0 | 0.113 | 0.042 |

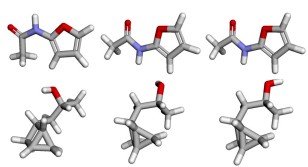

Figure 3: Molecular conformers generated by ETFLOW (left), +[OrbDiff] (center), and ground-truth (right).

Table 2 shows our method consistently improves both precision and diversity. OrbDiff achieves the best recall scores, including a 4% improvement in mean Cov@0.1, and the lowest Recall AMR (mean and median). It also maintains competitive precision at 0.1 Å. While MCF performs better at 0.5 Å precision, OrbDiff achieves the lowest AMR overall. Further experimental details and comparisons with more baselines are in Appendix C.1.

## 4.3 Crystal Structure Prediction (CSP)

CSP involves recovering 3D atomic positions and lattice parameters from chemical composition. Due to periodicity, it suffices to predict the structure within a single unit cell, where coordinates lie in the fractional domain $[0, 1)^{3 \times M}$. To handle this, DiffCSP uses a Wrapped Normal diffusion that respects periodic translation symmetry. We integrate Orbit Diffusion into DiffCSP, demonstrating that our approach extends beyond Gaussian diffusion models. We test two variants: OrbDiff_U, which samples uniformly over the translation group, and OrbDiff_WN, a time-dependent Wrapped Normal centered at zero, concentrating around $x_0$ at low noise and spreading out at high noise. Details of the OrbDiff_WN proposal are in Appendix C.2.3.

We evaluate our method on two CSP benchmarks: Perov-5 (Castelli et al., 2012a,b) and MP-20 (Jain et al., 2013). We use the three strongest baselines from the DiffCSP paper (Jiao et al., 2023): P-cG-SchNet (Gebauer et al., 2022), CDVAE (Xie et al., 2022), and DiffCSP, all with publicly available implementations. We evaluate performance using two standard metrics: Match Rate (the proportion of correctly matched structures in the test set) and RMSD (the average atomic deviation for matched samples, normalized by lattice volume). Full metric definitions and details are in Appendix C.2.4.

We further consider a relevant task, introduced by TGDMat (DAS et al., 2025), where crystal structures are generated conditioned on additional text descriptions of the desired structures. In this task, two types of descriptions are considered: long and short, with the latter being easier to obtain than the former. We follow the same evaluation framework as for Non-text-guided CSP.

Table 3: Text-guided CSP with TGDMat.

| Method | Perov-5 | | MP-20 | |
|---|---|---|---|---|
| | Match (↑) | RMSE (↓) | Match (↑) | RMSE (↓) |
| TGDMat (S) | 59.39 | 0.066 | 59.90 | 0.078 |
| +[OrbDiff_U] | 63.51 | 0.062 | 56.50 | 0.085 |
| +[OrbDiff_WN] | 65.57 | 0.054 | 61.29 | 0.072 |
| TGDMat (L) | 95.17 | 0.013 | 61.91 | 0.081 |
| +[OrbDiff_U] | 95.88 | 0.012 | 65.94 | 0.069 |
| +[OrbDiff_WN] | 95.98 | 0.012 | 66.74 | 0.069 |

Table 4: Crystal Structure Prediction (CSP).

| Method | Perov-5 | | MP-20 | |
|---|---|---|---|---|
| | Match (↑) | RMSE (↓) | Match (↑) | RMSE (↓) |
| P-cG-SchNet | 48.22 | 0.418 | 15.39 | 0.376 |
| CDVAE | 45.31 | 0.114 | 33.90 | 0.105 |
| DiffCSP | 52.02 | 0.076 | 51.49 | 0.063 |
| +[OrbDiff_U] | 52.29 | 0.078 | 54.47 | 0.054 |
| +[OrbDiff_WN] | 52.39 | 0.069 | 55.70 | 0.053 |

From Tables 3 and 4, one can see OrbDiff_WN consistently enhances the performance in all cases, with a notable increase from 59.39% to 65.57% for TGDmat (S) on Perov-5 and from 61.91% to 66.74% for TGDMat (L) on MP-20. At the same time, OrbDiff_U outperforms the baselines in 5 out of 6 cases, showing consistent benefits. Improvements are also observed consistently in RMSE.

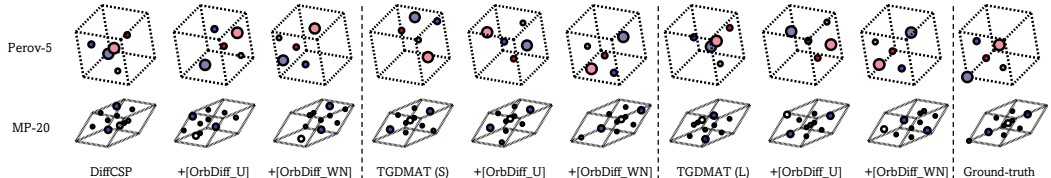

Figure 4: Qualitative comparison of Crystal Structure Predictions by 9 models, including DiffCSP, TGDMat (S) and TGDmat (L) with baselines, `OrbDiff_U`, and `OrbDiff_WN` against ground-truth samples on randomly selected samples from Perov-5 and MP-20 dataset.

## 4.4 Protein Structure Generation with PROTEINA

Table 5: Protein Structure Generation. Full comparisons are in the appendix. "+ [finetune]" denotes $\mathcal{M}_{FS}^{\text{no-tri}}$ finetuned with the original loss; "+ [OrbDiff]" uses `OrbDiff` for finetuning. Full table can be found in Appendix C.3.4

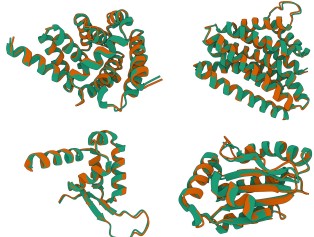

| Model | Designability | | Diversity | | Novelty | |
|---|---|---|---|---|---|---|
| | Fraction (↑) | scRMSD (↓) | Cluster (↑) | TM-score (↓) | PDB (↓) | AFDB (↓) |
| FoldFlow (OT) | 97.2 | - | 0.37 | 0.41 | 0.71 | 0.75 |
| $\mathcal{M}_{21M}$ | **99.0** | **0.72** | 0.30 | 0.39 | 0.81 | 0.84 |
| $\mathcal{M}_{FS}^{\text{no-tri}}$ | 93.8 | 1.04 | **0.62** | **0.36** | **0.69** | **0.76** |
| + [finetune] | 93.8 | 1.00 | 0.54 | 0.37 | 0.74 | 0.83 |
| + [OrbDiff] | **95.6** | **0.93** | 0.52 | 0.37 | 0.74 | 0.83 |

Figure 5: `OrbDiff` (orange) generated structures versus reference structure (green).

We adopt PROTEINA (denoted $\mathcal{M}$) (Geffner et al., 2025), the state-of-the-art model for protein structure generation, as a backbone. Although PROTEINA is a non-equivariant transformer, it performs well through extensive data augmentation. We finetune the 200M-parameter version of PROTEINA—$\mathcal{M}_{FS}^{\text{no-tri}}$—using `OrbDiff`, applying Rao-Blackwellization with a uniform proposal distribution over $SO(3)$, sampling 10,000 group elements.

For comparison, we also include the best-performing equivariant baseline, FoldFlow (OT) (Bose et al., 2024). We evaluate protein structures using three metrics: Designability (the feasibility of synthesizing the generated structures), Diversity, and Novelty. Designability is the most critical, while Diversity and Novelty are based on the designable samples, ensuring that the generated structures are synthetically plausible, diverse, and novel. Evaluation metric details are in Appendix C.3.3.

Table 5 shows that Orbit Diffusion boosts designability of $\mathcal{M}_{FS}^{\text{no-tri}}$ to 95.6% and lowers scRMSD to 0.93, outperforming naive finetuning while maintaining competitive diversity (Cluster: 0.52) and novelty (PDB: 0.74). The state-of-the-art $\mathcal{M}_{21M}$ (400M parameters) achieves higher designability (99.0%) and scRMSD (0.72) but at the cost of much lower diversity (Cluster: 0.30) and novelty (PDB: 0.81), showing a trade-off between validity and structural variety.

## 4.5 Benefits of OrbDiff: Efficiency, Stability, and Equivariance

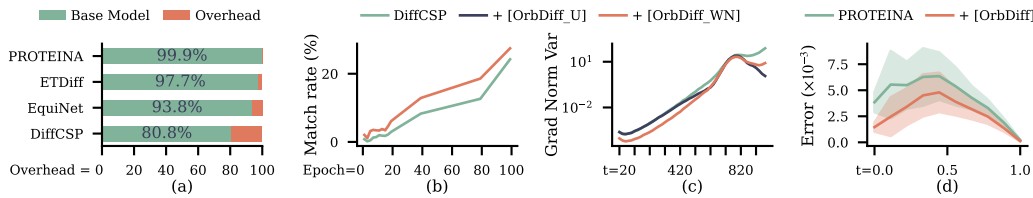

Figure 6: (a) `OrbDiff` introduces minimal computational overhead. (b) Match rate (↑) during DiffCSP training on the Perov-5 dataset: `OrbDiff` accelerates convergence. (c) `OrbDiff` reduces gradient variance across noise levels. (d) `OrbDiff` improves the equivariance of PROTEINA.

Adding `OrbDiff` introduces minimal overhead (Figure 6a)—about 20% for smaller models such as DiffCSP, and only 0.1% for larger ones like PROTEINA (200M parameters). In a 24-hour training run, this corresponds to just 1.4 additional minutes. In practice, the extra memory and computation for sampling and averaging over group elements are negligible compared to the overall training cost. For example, in our PROTEINA experiments—one of the largest protein diffusion models—we use 10,000 SO(3) samples per training example, which adds only $\sim$40 MB per GPU to the total memory usage of $\sim$54 GB.

To better understand how `OrbDiff` accelerates convergence, we compare DiffCSP and DiffCSP + [`OrbDiff`] across training checkpoints. As shown in Figure 6b, integrating `OrbDiff` consistently improves match rates throughout training, especially in the early stages.

We also compare the empirical gradient norm variance of DiffCSP (Perov-5) with `OrbDiff_U` and `OrbDiff_WN` across the full training set and various timesteps. As expected, `OrbDiff_WN` achieves significantly lower variance at small to intermediate noise levels by sampling locally around $x_0$, while `OrbDiff_U` performs better at high noise levels due to its more global sampling. Both methods substantially reduce variance compared to DiffCSP. Finally, to assess equivariance preservation, we compare PROTEINA + [finetune] and PROTEINA + [`OrbDiff`] using an equivariance test from (Geffner et al., 2025):

$$\text{Error}_t = \mathbb{E}_{x \sim \hat{q}^G(x_0),\ x_t \sim \hat{q}_t^G(x_t|x_0),\ g \sim \text{Unif}(SO(3))} \left[ \text{RMSD}(g \circ \phi(x_t, t),\ \phi(g \circ x_t, t)) \right]. \tag{13}$$

As shown in Figure 6d, `OrbDiff` substantially reduces equivariance error compared to naive finetuning, indicating improved geometric consistency in the model's denoising.

## 5  Related Work

Equivariant neural networks have been extensively studied for their ability to encode symmetry priors in domains such as vision (Cohen & Welling, 2016; Worrall et al., 2017), 3D geometry (Thomas et al., 2018; Deng et al., 2021), and molecular modeling (Le et al., 2022; Kozinsky et al., 2023). More recently, these ideas have been incorporated into diffusion models to improve generative performance in structured domains like proteins and molecules (Corso et al., 2024; Hoogeboom et al., 2022; Igashov et al., 2024). By designing the denoising network to be equivariant under a symmetry group, these models better align with the underlying data distribution. A comprehensive discussion of related work is provided in Appendix A.

## 6  Conclusion

Orbit Diffusion is a framework for training generative models under symmetry constraints by reducing gradient variance through Rao–Blackwellization. The approach unifies equivariant architectures and data augmentation strategies within a single probabilistic formulation, providing a provably lower-variance estimator while maintaining computational efficiency. Theoretically, we show that the proposed loss admits equivariant minimizers and connects to existing score-based and diffusion formulations. Empirically, Orbit Diffusion demonstrates strong and consistent performance across a diverse set of generative tasks, including molecular, crystal structure, and protein structure modeling. By bridging the gap between symmetry-aware modeling and optimization stability, our method improves both the scalability and practical applicability of equivariant generative models in scientific domains.

## 7  Limitations

To estimate the conditional expectation in our gradient estimator, we employ orbit sampling to reduce variance, which leads to improved performance. While efficient, our sampling scheme inherently constrains the choice of proposal distributions and may limit modeling flexibility, especially for tasks requiring diverse or adaptive noise structures. Furthermore, the number of orbit samples directly affects both computational cost and estimation accuracy, suggesting that dynamically adjusting the sampling strategy during training could yield better efficiency–accuracy trade-offs. Future work could explore adaptive or learned proposal mechanisms to enhance generalization and robustness across broader data regimes.

## Acknowledgements

This work was supported in part by the Deutsche Forschungsgemeinschaft (DFG, German Research Foundation) under Germany's Excellence Strategy – EXC 2075 – 390740016; the DARPA ANSR program under award FA8750-23-2-0004; the DARPA CODORD program under award HR00112590089; the NSF grant IIS-1943641; the National University of Singapore Start-up Grant (Award No. SUG-251RES250); and gifts from Adobe Research, Cisco Research, and Amazon. We also acknowledge the support of the Stuttgart Center for Simulation Science (SimTech). VT and MN thank the International Max Planck Research School for Intelligent Systems (IMPRS-IS) for support. This work further received partial support from the Diabetes Center Berne and from the Ministry of Science, Research and the Arts Baden-Württemberg through the Artificial Intelligence Software Academy (AISA).

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

# RAO-BLACKWELL GRADIENT ESTIMATORS FOR EQUIVARIANT DENOISING DIFFUSION
### ADDITIONAL MATERIAL

## A   Related Work

**Equivariant Neural Networks.**   Equivariant neural networks have attracted significant attention for tasks involving structured data, such as computer vision (Cohen & Welling, 2016; Worrall et al., 2017), 3D modeling (Thomas et al., 2018; Fuchs et al., 2020; Deng et al., 2021), quantum mechanics and quantum field theory (Gerdes et al., 2023; Hermann et al., 2020), biomolecular design (Le et al., 2022; Kozinsky et al., 2023; Geffner et al., 2025). These networks exploit group symmetries to ensure consistent outputs under transformations such as rotations, translations, and permutations, which are commonly encountered in many scientific domains. By incorporating symmetric inductive biases, equivariant networks enhance model generalization and reduce data requirements by naturally encoding symmetry constraints. However, challenges remain, such as high computational complexity (He et al., 2021) and difficulties in effectively learning with stochastic gradient descent (SGD) (Abbe & Boix-Adserà, 2022).

**Equivariance and Diffusion Models.**   Diffusion models have become a dominant class of generative models, known for their effectiveness in modeling complex data distributions (Song et al., 2021a; Ho et al., 2020; Karras et al., 2022). They work by gradually adding noise to data in a forward process and learning to reverse this corruption using a denoising network $\phi_\theta$. Incorporating symmetry, particularly equivariance, into these models has shown significant benefits in domains where data lies on geometric manifolds with known symmetries—such as structural biology (Corso et al., 2024; Yim et al., 2023b; Schneuing et al., 2024; Igashov et al., 2024), molecular modeling (Hoogeboom et al., 2022; Guan et al., 2023; Le et al., 2024), and material design (Xie et al., 2022; Gebauer et al., 2022; Jiao et al., 2023). In such settings, where symmetries are typically governed by the Euclidean group or its subgroups, enforcing equivariance in the generative process helps ensure physical plausibility and improves generalization. A common approach is to design the denoiser $\phi_\theta$ to be equivariant under a symmetry group $G$, which encourages the learned distribution to converge toward the correct invariant target (Hoogeboom et al., 2022; Xu et al., 2022a; Bose et al., 2024; Chen et al., 2024).

Nonetheless, recent advances have shown that models without explicit equivariant constraints can still achieve strong empirical performance, thanks to greater flexibility in architectural design and effective use of data augmentation. These approaches may implicitly capture symmetry through training strategies rather than architectural bias, as demonstrated by state-of-the-art models such as PROTEINA (Geffner et al., 2025) and AlphaFold 3 (Abramson et al., 2024a).

## B   Theoretical Proofs

### B.1   Proof of Theorem 1

Variance Reduction of Rao-Blackwell estimator.

**Theorem 1.** *Let $\widehat{\nabla}_\phi[\mathcal{L}_t^G(\phi)]$ and $\widehat{\nabla}_\phi^{(RB)}[\mathcal{L}_t^G(\phi)]$ denote the gradient estimators of Equation (4) and Equation (5), respectively. Suppose we can compute $\mathbb{E}_{x_0 \sim \hat{q}_t^G(x_0|x_t)}[x_0]$. Then*

$$\text{Var}\left(\widehat{\nabla}_\phi^{(RB)}[\mathcal{L}_t^G(\phi)]\right) \le \text{Var}\left(\widehat{\nabla}_\phi[\mathcal{L}_t^G(\phi)]\right).$$

*Moreover, the inequality is* strict *unless $\hat{q}_t^G(x_0 \mid x_t)$ is a Dirac delta, which is rarely the case in generative modeling where $x_0$ is typically stochastic given $x_t$.*

*Proof.* By definition of the Rao-Blackwellized estimator, we have

$$\widehat{\nabla}_\phi^{(RB)}[\mathcal{L}_t^G(\phi)] = \mathbb{E}_{x_0 \sim \hat{q}_t^G(x_0|x_t)}\left[\widehat{\nabla}_\phi[\mathcal{L}_t^G(\phi)] \mid x_t\right].$$

Thus, the Rao-Blackwell theorem implies that conditioning reduces variance:

$$\text{Var}\left(\mathbb{E}\left[\widehat{\nabla}_\phi[\mathcal{L}_t^G(\phi)] \mid x_t\right]\right) \le \text{Var}\left(\widehat{\nabla}_\phi[\mathcal{L}_t^G(\phi)]\right),$$

with equality if and only if $\widehat{\nabla}_\phi[\mathcal{L}_t^G(\phi)]$ is almost surely a function of $x_t$ (i.e., deterministic given $x_t$).

To verify this explicitly, consider the variance of the Rao-Blackwellized estimator:

$$\text{Var}\left(\widehat{\nabla}_\phi^{(RB)}[\mathcal{L}_t^G(\phi)]\right) = \mathbb{E}_{x_t}\left[\left(\mathbb{E}\left[\widehat{\nabla}_\phi[\mathcal{L}_t^G(\phi)] \mid x_t\right] - \nabla_\phi[\mathcal{L}_t^G(\phi)]\right)^2\right]$$

$$= \mathbb{E}_{x_t}\left[\left(\mathbb{E}\left[\widehat{\nabla}_\phi[\mathcal{L}_t^G(\phi)] - \nabla_\phi[\mathcal{L}_t^G(\phi)] \mid x_t\right]\right)^2\right]$$

$$\leq \mathbb{E}_{x_t}\left[\mathbb{E}\left[\left(\widehat{\nabla}_\phi[\mathcal{L}_t^G(\phi)] - \nabla_\phi[\mathcal{L}_t^G(\phi)]\right)^2 \mid x_t\right]\right]$$

$$= \text{Var}\left(\widehat{\nabla}_\phi[\mathcal{L}_t^G(\phi)]\right),$$

where the inequality follows from Jensen's inequality (applied to the convex function $f(z) = z^2$).

Therefore, the Rao-Blackwellized estimator has variance less than or equal to the original estimator, and strictly less unless the conditional variance given $x_t$ is zero. □

## B.2 Equivalence of Symmetrized and Equivariant Diffusion Losses

**Proposition 1.** *Let $\mathcal{L}_t^{(1)}(\phi)$ denote the* Symmetrized Diffusion Loss, *defined as*

$$\mathcal{L}_t^{(1)}(\phi) = \mathbb{E}_{x_0' \sim \hat{q}^G(x_0)} \mathbb{E}_{x_t \sim \hat{q}_t^G(x_t|x_0')}\left[\left\|\phi(x_t, t) - \mathbb{E}_{x_0 \sim \hat{q}^G(x_0|x_t)}[x_0]\right\|^2\right],$$

*and let $\mathcal{L}_t^{(2)}(\phi)$ denote the corresponding loss defined on the original (non-symmetrized) data distribution $\hat{q}(x_0)$:*

$$\mathcal{L}_t^{(2)}(\phi) = \mathbb{E}_{x_0' \sim \hat{q}(x_0)} \mathbb{E}_{x_t \sim \hat{q}_t(x_t|x_0')}\left[\left\|\phi(x_t, t) - \mathbb{E}_{x_0 \sim \hat{q}^G(x_0|x_t)}[x_0]\right\|^2\right].$$

*Suppose $\phi(x_t, t)$ is a $G$-equivariant function, i.e.,*

$$\phi(g \circ x_t, t) = g \circ \phi(x_t, t) \quad \forall g \in G, \ x_t \in \Omega.$$

*Then $\mathcal{L}_t^{(1)}(\phi)$ and $\mathcal{L}_t^{(2)}(\phi)$ are equivalent in the sense that they share the same minimizer and gradient with respect to $\phi$.*

To prove this, we need the following lemma:

**Symmetrized Forward Diffusion Distributions**

**Lemma 1.** *Let $\hat{q}(x_0)$ be an empirical distribution and $\hat{q}^G(x_0)$ its symmetrized counterpart under a symmetry group $G$, defined by $\hat{q}^G(x_0) = S_G[\hat{q}](x_0)$. Suppose a forward diffusion process acting on $\hat{q}(x_0)$ yields time-dependent marginal distributions $\hat{q}_t(x_t)$. Let a similar process act on $\hat{q}^G(x_0)$ to generate $\hat{q}_t^G(x_t)$. Then, for all $t \geq 0$, the following holds:*

$$\hat{q}_t^G(x_t) = S_G[\hat{q}_t](x_t).$$

The proof can be found in Appendix B.3.

We also need the following lemma:

**Lemma 2.** *Let $G$ be a group of isometries acting on $\Omega$, and suppose the distribution $\hat{q}_t^G(x_t \mid x_0)$ is equivariant under the action of $G$. Then for any $g \in G$, the following identity holds:*

$$\mathbb{E}_{x_0 \sim \hat{q}_t^G(x_0|g \circ x_t)}[x_0] = g \circ \mathbb{E}_{x_0 \sim \hat{q}_t^G(x_0|x_t)}[x_0].$$

*Proof.* We start by expanding the conditional expectation:

$$\mathbb{E}_{x_0 \sim \hat{q}_t^G(x_0 | g \circ x_t)}[x_0] = \int_\Omega x_0 \, \hat{q}_t^G(x_0 \mid g \circ x_t) \, dx_0$$

$$= \int_\Omega x_0 \, \hat{q}_t^G(g \circ x_t \mid x_0) \frac{\hat{q}^G(x_0)}{\hat{q}_t^G(g \circ x_t)} \, dx_0.$$

Now, apply the change of variable $x_0 = g \circ \bar{x}_0$. Since $g$ is an isometry, the Lebesgue measure is invariant, i.e., $dx_0 = d\bar{x}_0$. Therefore:

$$= \int_\Omega (g \circ \bar{x}_0) \, \hat{q}_t^G(g \circ x_t \mid g \circ \bar{x}_0) \frac{\hat{q}^G(g \circ \bar{x}_0)}{\hat{q}_t^G(g \circ x_t)} \, d\bar{x}_0$$

$$= g \circ \int_\Omega \bar{x}_0 \, \hat{q}_t^G(x_t \mid \bar{x}_0) \frac{\hat{q}^G(\bar{x}_0)}{\hat{q}_t^G(x_t)} \, d\bar{x}_0$$

$$= g \circ \mathbb{E}_{x_0 \sim \hat{q}_t^G(x_0 | x_t)}[x_0],$$

where the second equality follows from the equivariance of $\hat{q}_t^G$ and the invariance of $\hat{q}^G$ under the group action. This concludes the proof. $\square$

Now the prove the Proposition 1.

*Proof.* We begin by simplifying $\mathcal{L}_t^{(1)}$:

$$\mathcal{L}_t^{(1)}(\phi) = \mathbb{E}_{x_t \sim \hat{q}_t^G(x_t)} \mathbb{E}_{x_0' \sim \hat{q}^G(x_0 | x_t)} \left[ \left\| \phi(x_t, t) - \mathbb{E}_{x_0 \sim \hat{q}_t^G(x_0 | x_t)}[x_0] \right\|^2 \right]$$

$$= \mathbb{E}_{x_t \sim \hat{q}_t^G(x_t)} \left[ \left\| \phi(x_t, t) - \mathbb{E}_{x_0 \sim \hat{q}_t^G(x_0 | x_t)}[x_0] \right\|^2 \right].$$

Similarly,

$$\mathcal{L}_t^{(2)}(\phi) = \mathbb{E}_{x_t \sim \hat{q}_t(x_t)} \left[ \left\| \phi(x_t, t) - \mathbb{E}_{x_0 \sim \hat{q}_t^G(x_0 | x_t)}[x_0] \right\|^2 \right].$$

We now rewrite $\mathcal{L}_t^{(1)}$ in integral form:

$$\mathcal{L}_t^{(1)}(\phi) = \int_\Omega \left\| \phi(x_t, t) - \mathbb{E}_{x_0 \sim \hat{q}_t^G(x_0 | x_t)}[x_0] \right\|^2 \hat{q}_t^G(x_t) \, dx_t.$$

Using the decomposition of the measure over the group action, we change variables:

$$\int_\Omega f(x_t) \, dx_t = \int_{\Omega/G} \int_G f(g \circ x_t) \, d\mu_G(g) \, dx_t.$$

Thus,

$$\mathcal{L}_t^{(1)}(\phi) = \int_{\Omega/G} \int_G \left\| \phi(g \circ x_t, t) - \mathbb{E}_{x_0 \sim \hat{q}_t^G(x_0 | g \circ x_t)}[x_0] \right\|^2 \hat{q}_t^G(g \circ x_t) \, d\mu_G(g) \, dx_t$$

$$= \int_{\Omega/G} \int_G \left\| g \circ \phi(x_t, t) - \mathbb{E}_{x_0 \sim \hat{q}_t^G(x_0 | g \circ x_t)}[x_0] \right\|^2 \hat{q}_t^G(x_t) \, d\mu_G(g) \, dx_t,$$

where we used the $G$-equivariance of $\phi$ and the invariance of $\hat{q}_t^G$.

Since $G$ acts isometrically, we apply:

$$\left\| g \circ \phi(x_t, t) - \mathbb{E}_{x_0 \sim \hat{q}_t^G(x_0 | g \circ x_t)}[x_0] \right\|^2 = \left\| \phi(x_t, t) - g^{-1} \circ \mathbb{E}_{x_0 \sim \hat{q}_t^G(x_0 | g \circ x_t)}[x_0] \right\|^2.$$

Next, applying Lemma 2, we have:

$$\left\| g \circ \phi(x_t, t) - \mathbb{E}_{x_0 \sim \hat{q}_t^G(x_0 | g \circ x_t)}[x_0] \right\|^2 = \left\| \phi(x_t, t) - \mathbb{E}_{x_0 \sim \hat{q}_t^G(x_0 | x_t)}[x_0] \right\|^2$$

We conclude:

$$\mathcal{L}_t^{(1)}(\phi) = \int_{\Omega/G} \int_G \left\| \phi(x_t, t) - \mathbb{E}_{x_0 \sim \hat{q}_t^G(x_0|x_t)}[x_0] \right\|^2 \hat{q}_t^G(x_t) \, \mathrm{d}\mu_G(g) \, \mathrm{d}x_t$$

$$= \int_{\Omega/G} \left\| \phi(x_t, t) - \mathbb{E}_{x_0 \sim \hat{q}_t^G(x_0|x_t)}[x_0] \right\|^2 \hat{q}_t^G(x_t) \, \mathrm{d}x_t.$$

Next, we similarly transform $\mathcal{L}_t^{(2)}(\phi)$:

$$\mathcal{L}_t^{(2)}(\phi) = \int_{\Omega/G} \int_G \left\| \phi(g \circ x_t, t) - \mathbb{E}_{x_0 \sim \hat{q}_t^G(x_0|g \circ x_t)}[x_0] \right\|^2 \hat{q}_t(g \circ x_t) \, \mathrm{d}\mu_G(g) \, \mathrm{d}x_t$$

$$= \int_{\Omega/G} \left\| \phi(x_t, t) - \mathbb{E}_{x_0 \sim \hat{q}_t^G(x_0|x_t)}[x_0] \right\|^2 \left( \int_G \hat{q}_t(g \circ x_t) \, \mathrm{d}\mu_G(g) \right) \mathrm{d}x_t.$$

Using the theoretical result from Appendix B.3, we have:

$$\hat{q}_t^G(x_t) = S_G[\hat{q}_t](x_t) = \int_G \hat{q}_t(g \circ x_t) \, \mathrm{d}\mu_G(g),$$

we conclude:

$$\mathcal{L}_t^{(2)}(\phi) = \int_{\Omega/G} \left\| \phi(x_t, t) - \mathbb{E}_{x_0 \sim \hat{q}_t^G(x_0|x_t)}[x_0] \right\|^2 \hat{q}_t^G(x_t) \, \mathrm{d}x_t = \mathcal{L}_t^{(1)}(\phi).$$

Hence, the two loss functions are equivalent in the sense that they yield the same gradients and minimizers with respect to $\phi$. $\qquad\square$

## B.3 Symmetrized Forward Diffusion Distribution.

Below is the formal lemma and the proof for the symmetrized forward diffusion distribution.

---

**Symmetrized Forward Diffusion Distributions**

**Lemma 1.** *Let $\hat{q}(x_0)$ be an empirical distribution and $\hat{q}^G(x_0)$ its symmetrized counterpart under a symmetry group $G$, defined by $\hat{q}^G(x_0) = S_G[\hat{q}](x_0)$. Suppose a forward diffusion process acting on $\hat{q}(x_0)$ yields time-dependent marginal distributions $\hat{q}_t(x_t)$. Let a similar process act on $\hat{q}^G(x_0)$ to generate $\hat{q}_t^G(x_t)$. Then, for all $t \geq 0$, the following holds:*

$$\hat{q}_t^G(x_t) = S_G[\hat{q}_t](x_t).$$

---

*Proof.* The marginal distribution at time step $t$ of a diffusion process is defined as

$$\hat{q}_t(x_t) = \int_\Omega q_t(x_t \mid x_0)\hat{q}(x_0) \, \mathrm{d}x_0,$$

where $q_t(x_t \mid x_0) = \mathcal{N}(x_t; \alpha_t x_0, \sigma_t^2 I)$ is the Gaussian diffusion kernel, which is equivariant under isometry group transformations. Specifically, for any $g \in G$, we have

$$q_t(x_t \mid x_0) = q_t(g \circ x_t \mid g \circ x_0).$$

The symmetrized marginal distribution at time $t$ is defined as

$$S_G[\hat{q}_t](x_t) = \int_G \hat{q}_t(g \circ x_t) \mathrm{d}\mu_G(g)$$

$$= \int_G \left[ \int_\Omega q_t(g \circ x_t \mid x_0)\hat{q}(x_0) \, \mathrm{d}x_0 \right] \mathrm{d}\mu_G(g)$$

$$= \int_G \int_\Omega q_t(g \circ x_t \mid x_0)\hat{q}(x_0) \, \mathrm{d}x_0 \mathrm{d}\mu_G(g).$$

Next, we compute the marginal distribution at time $t$ when the forward process is applied to the symmetrized data distribution:

$$\hat{q}_t^G(x_t) = \int_\Omega q_t(x_t \mid x_0) S_G[\hat{q}](x_0) \, \mathrm{d}x_0$$

$$= \int_\Omega q_t(x_t \mid x_0) \int_G \hat{q}(g \circ x_0) \mathrm{d}\mu_G(g) \mathrm{d}x_0$$

$$= \int_\Omega \int_G q_t(x_t \mid x_0) \hat{q}(g \circ x_0) \mathrm{d}\mu_G(g) \mathrm{d}x_0$$

$$= \int_G \int_\Omega q_t(x_t \mid x_0) \hat{q}(g \circ x_0) \mathrm{d}x_0 \mathrm{d}\mu_G(g).$$

Applying a change of variable $x_0 \mapsto g^{-1} \circ x_0$, we get

$$\hat{q}_t^G(x_t) = \int_G \int_\Omega q_t(x_t \mid g^{-1} \circ x_0) \hat{q}(g \circ [g^{-1} \circ x_0]) \mathrm{d}(g^{-1} \circ x_0) \mathrm{d}\mu_G(g)$$

$$= \int_G \int_\Omega q_t(x_t \mid g^{-1} \circ x_0) \hat{q}(x_0) \mathrm{d}x_0 \mathrm{d}\mu_G(g) \quad \text{(since the Jacobian of $g$ is 1)}$$

$$= \int_G \int_\Omega q_t(g \circ x_t \mid x_0) \hat{q}(x_0) \mathrm{d}x_0 \mathrm{d}\mu_G(g) \quad \text{(by kernel equivariance)}.$$

Thus, we have

$$\hat{q}_t^G(x_t) = S_G[\hat{q}_t](x_t).$$

This completes the proof. $\qquad\square$

## B.4    Proof of Theorem 2

> **Unbiased gradient and equivariance of `OrbDiff`**
>
> **Theorem 2.** *Let $G$ be a locally compact isometry group acting on data space $\Omega$, and suppose the forward kernels $\hat{q}_t^G(x_t \mid x_0)$ are $G$-invariant: $\hat{q}_t^G(g \circ x_t \mid g \circ x_0) = \hat{q}_t^G(x_t \mid x_0)$ for all $g \in G$. Then:*
>
> 1. *The `OrbDiff` target $\phi^*(x_0, x_t, t)$ satisfies: $\phi^*(x_0, h \circ x_t, t) = h \circ \phi^*(x_0, x_t, t)$ for all $h \in G$.*
> 2. *The gradient of the `OrbDiff` loss (12) equals that of the ideal loss (5), i.e., `OrbDiff` provides an unbiased gradient estimator.*

**We first prove that the `OrbDiff` target is equivariant:**

*Proof.* We compute $\phi^*(x_0, h \circ x_t, t)$ using the definition:

$$\phi^*(x_0, h \circ x_t, t) = \frac{1}{Z(h \circ x_t, x_0)} \int_G (g \circ x_0) \, \hat{q}_t^G(h \circ x_t \mid g \circ x_0) \, \mathrm{d}\mu_G(g).$$

By the equivariance of $\hat{q}_t^G$, we have:

$$\hat{q}_t^G(h \circ x_t \mid g \circ x_0) = \hat{q}_t^G(x_t \mid h^{-1} \circ g \circ x_0).$$

Letting $g' = h^{-1} \circ g$, so $g = h \circ g'$, and using the left-invariance of the Haar measure $\mu_G$, we get:

$$\phi^*(x_0, h \circ x_t, t) = \frac{1}{Z(h \circ x_t, x_0)} \int_G (h \circ g' \circ x_0) \, \hat{q}_t^G(x_t \mid g' \circ x_0) \, \mathrm{d}\mu_G(g').$$

Factoring out $h$ from the integrand gives:

$$= h \circ \left[ \frac{1}{Z(h \circ x_t, x_0)} \int_G (g' \circ x_0) \, \hat{q}_t^G(x_t \mid g' \circ x_0) \, \mathrm{d}\mu_G(g') \right].$$

It remains to show that $Z(h \circ x_t, x_0) = Z(x_t, x_0)$, where:

$$Z(x_t, x_0) := \int_G \hat{q}_t^G(x_t \mid g \circ x_0) \, d\mu_G(g).$$

Using the same substitution:

$$Z(h \circ x_t, x_0) = \int_G \hat{q}_t^G(h \circ x_t \mid g \circ x_0) \, d\mu_G(g) = \int_G \hat{q}_t^G(x_t \mid h^{-1} \circ g \circ x_0) \, d\mu_G(g) = \int_G \hat{q}_t^G(x_t \mid g' \circ x_0) \, d\mu_G(g') = Z(x_t).$$

Therefore,

$$\phi^*(x_0, h \circ x_t, t) = h \circ \phi^*(x_0, x_t, t).$$

$\square$

**Next, we prove that `OrbDiff` yields an unbiased gradient estimator:**

*Proof.* Our goal is to estimate the following gradient:

$$\nabla_\phi \mathcal{L}_t^G(\phi) = 2\mathbb{E}_{x_t \sim \hat{q}_t^G(x_t)} \left[ \phi(x_t, t) - \mathbb{E}_{x_0 \sim \hat{q}_t^G(x_0|x_t)}[x_0] \right]. \tag{14}$$

Our proposed Rao-Blackwell loss function has the same gradient as $\mathcal{L}_t^G(\phi)$, but with reduced variance due to the use of the conditional expectation $\mathbb{E}_{x_0 \sim \hat{q}_t^G(x_0|x_t)}[x_0]$:

$$\mathcal{L}_t^{\text{RB}}(\phi) = \mathbb{E}_{x_0' \sim \hat{q}^G(x_0')} \mathbb{E}_{x_t \sim \hat{q}_t^G(x_t|x_0')} \left[ \left\| \phi(x_t, t) - \mathbb{E}_{x_0 \sim \hat{q}_t^G(x_0|x_t)}[x_0] \right\|^2 \right]. \tag{15}$$

However, computing $\mathbb{E}_{x_0 \sim \hat{q}_t^G(x_0|x_t)}[x_0]$ can be computationally expensive. To address this, we introduce OrbDiff, which uses a biased proposal distribution to approximate this expectation using only samples from the orbit of the $x_0$ that generated $x_t$. This yields the alternative target:

$$\phi^*(x_0, x_t, t) = \frac{1}{Z(x_0, x_t)} \int_G (g \circ x_0) \, \hat{q}_t^G(g \circ x_0 \mid x_t) \, d\mu_G(g), \tag{16}$$

where the normalization constant is

$$Z(x_0, x_t) = \int_G \hat{q}_t^G(g \circ x_0 \mid x_t) \, d\mu_G(g). \tag{17}$$

This matches Equation (11), which can be verified via straightforward transformations.

Although $\phi^*$ differs from $\mathbb{E}_{x_0 \sim \hat{q}_t^G(x_0|x_t)}[x_0]$, we show that replacing the Rao-Blackwell target in Eq. (15) with $\phi^*$ yields a loss whose gradient still matches the original gradient. This relies on the assumption that the forward conditional distribution is equivariant under the group action, i.e.,

$$\hat{q}_t^G(g \circ x_t \mid g \circ x_0) = \hat{q}_t^G(x_t \mid x_0), \quad \forall g \in G, \tag{18}$$

which is a natural condition satisfied in many generative models such as diffusion models or flow matching with isotropic Gaussian priors.

Under this assumption, the OrbDiff loss is given by:

$$\mathcal{L}_t^{\text{OrbDiff}}(\phi) = \mathbb{E}_{x_0 \sim \hat{q}^G(x_0)} \mathbb{E}_{x_t \sim \hat{q}_t^G(x_t|x_0)} \left[ \|\phi(x_t, t) - \phi^*(x_0, x_t, t)\|^2 \right] \tag{19}$$

$$= \mathbb{E}_{x_t \sim \hat{q}_t^G(x_t)} \mathbb{E}_{x_0 \sim \hat{q}_t^G(x_0|x_t)} \left[ \|\phi(x_t, t) - \phi^*(x_0, x_t, t)\|^2 \right]. \tag{20}$$

Taking the gradient with respect to $\phi$, we obtain:

$$\nabla_\phi \mathcal{L}_t^{\text{OrbDiff}}(\phi) = 2\mathbb{E}_{x_t \sim \hat{q}_t^G(x_t)} \mathbb{E}_{x_0 \sim \hat{q}_t^G(x_0|x_t)} \left[ \phi(x_t, t) - \phi^*(x_0, x_t, t) \right] \tag{21}$$

$$= 2\mathbb{E}_{x_t \sim \hat{q}_t^G(x_t)} \left[ \phi(x_t, t) - \mathbb{E}_{x_0 \sim \hat{q}_t^G(x_0|x_t)}[\phi^*(x_0, x_t, t)] \right]. \tag{22}$$

Thus, to ensure that OrbDiff yields the correct gradient, it suffices to show:

$$\mathbb{E}_{x_0 \sim \hat{q}_t^G(x_0|x_t)}[\phi^*(x_0, x_t, t)] = \mathbb{E}_{x_0 \sim \hat{q}_t^G(x_0|x_t)}[x_0]. \tag{23}$$

We compute:

$$\mathbb{E}_{x_0 \sim \hat{q}_t^G(x_0 | x_t)}[\phi^*(x_0, x_t, t)] = \int_\Omega \int_G \phi^*(g' \circ x_0, x_t, t) \, \hat{q}_t^G(g' \circ x_0 \mid x_t) \, \mathrm{d}\mu_G(g') \, \mathrm{d}x_0. \qquad (24)$$

Substituting the expression for $\phi^*$ and applying the change of variables $g \mapsto g \cdot g'$ with left-invariant Haar measure $\mu_G$, we have:

$$= \int_\Omega \int_G \left[ \frac{1}{Z(g' \circ x_0, x_t)} \int_G (g \circ [g' \circ x_0]) \, \hat{q}_t^G(g \circ [g' \circ x_0] \mid x_t) \, \mathrm{d}\mu_G(g) \right] \hat{q}_t^G(g' \circ x_0 \mid x_t) \, \mathrm{d}\mu_G(g') \, \mathrm{d}x_0 \tag{25}$$

$$= \int_\Omega \int_G \left[ \frac{1}{Z(x_0, x_t)} \int_G (g \circ x_0) \, \hat{q}_t^G(g \circ x_0 \mid x_t) \, \mathrm{d}\mu_G(g) \right] \hat{q}_t^G(g' \circ x_0 \mid x_t) \, \mathrm{d}\mu_G(g') \, \mathrm{d}x_0 \tag{26}$$

$$= \int_\Omega \left[ \frac{1}{Z(x_0, x_t)} \int_G (g \circ x_0) \, \hat{q}_t^G(g \circ x_0 \mid x_t) \, \mathrm{d}\mu_G(g) \right] \left[ \int_G \hat{q}_t^G(g' \circ x_0 \mid x_t) \, \mathrm{d}\mu_G(g') \right] \mathrm{d}x_0 \tag{27}$$

$$= \int_\Omega \left[ \frac{1}{Z(x_0, x_t)} \int_G (g \circ x_0) \, \hat{q}_t^G(g \circ x_0 \mid x_t) \, \mathrm{d}\mu_G(g) \right] Z(x_0, x_t) \, \mathrm{d}x_0 \tag{28}$$

$$= \int_\Omega \int_G (g \circ x_0) \, \hat{q}_t^G(g \circ x_0 \mid x_t) \, \mathrm{d}\mu_G(g) \, \mathrm{d}x_0 \tag{29}$$

$$= \mathbb{E}_{x_0 \sim \hat{q}_t^G(x_0 | x_t)}[x_0]. \tag{30}$$

Therefore, despite $\phi^*$ not being equal to the conditional expectation at each $x_t$, the gradient induced by the OrbDiff loss matches the desired gradient. OrbDiff thus provides an unbiased estimate of $\nabla_\phi \mathcal{L}_t^G(\phi)$, while using only samples from the orbit of $x_0$.

$\square$

## B.5 Unconstrained Non-Symmetrized Diffusion Minimizer is not guaranteed G-equivaraint

> **Lemma 2.** *Let*
> $$\mathcal{L}_t(\phi) = \mathbb{E}_{(x_0, x_t) \sim \hat{q}(x_0, x_t)} \left[ \| \phi(x_t, t) - x_0 \|^2 \right]$$
> *be the unconstrained Non-Symmetrized Diffusion loss, where $\hat{q}(x_0, x_t)$ is the empirical joint distribution of clean and noisy data. Then the minimizer $\phi^*(x_t, t)$ of $\mathcal{L}_t$ is the conditional expectation:*
> $$\phi^*(x_t, t) = \mathbb{E}_{x_0 \sim \hat{q}_t(x_0 | x_t)}[x_0].$$
> *However, this minimizer is not guaranteed to be equivariant under the action of a symmetry group $G$-equivariant.*

*Proof.* To find the minimizer of the diffusion loss, we first compute the stationary point. The loss is given by:

$$\mathcal{L}_t(\phi) = \mathbb{E}_{(x_0, x_t) \sim \hat{q}(x_0, x_t)} \left[ \| \phi(x_t, t) - x_0 \|^2 \right] = \mathbb{E}_{x_t \sim \hat{q}_t(x_t)} \mathbb{E}_{x_0 \sim \hat{q}_t(x_0 | x_t)} \left[ \| \phi(x_t, t) - x_0 \|^2 \right].$$

The gradient of the internal expectation is:

$$\nabla_\phi \mathbb{E}_{x_0 \sim \hat{q}_t(x_0 | x_t)} \left[ \| \phi(x_t, t) - x_0 \|^2 \right] = 2 \mathbb{E}_{x_0 \sim \hat{q}_t(x_0 | x_t)} \left[ \phi(x_t, t) - x_0 \right]$$
$$= 2 \left( \phi(x_t, t) - \mathbb{E}_{x_0 \sim \hat{q}_t(x_0 | x_t)}[x_0] \right).$$

Setting the gradient to zero gives the minimizer:

$$\phi^*(x_t, t) = \mathbb{E}_{x_0 \sim \hat{q}_t(x_0 | x_t)}[x_0] = \int_\Omega x_0 \hat{q}_t(x_0 \mid x_t) \, dx_0.$$

Next, we provide a counterexample to show that $\phi^*(x_t, t)$ is not guaranteed to be equivariant.

**Counterexample: Translation in 1D**

1. **Data**: Two points $x_0^1 = 0$ and $x_0^2 = 1$, with uniform empirical distribution $\hat{q}(x_0^i) = 0.5$.

2. **Group action**: Translation by $a = 1$, i.e., $g \circ x = x + 1$.

3. **Diffusion kernel**: $q_t(x_t \mid x_0) = \mathcal{N}(x_t; \alpha_t x_0, \sigma_t^2)$.

First, rewrite the minimizer:

$$\phi^*(x_t, t) = \sum_{i=1}^{N} x_0^i \hat{q}_t(x_0^i \mid x_t)$$

$$= \sum_{i=1}^{N} x_0^i \hat{q}_t(x_t \mid x_0^i) \frac{\hat{q}(x_0^i)}{\hat{q}_t(x_t)}$$

$$= \frac{1}{\hat{q}_t(x_t)} \sum_{i=1}^{N} x_0^i \hat{q}_t(x_t \mid x_0^i) \hat{q}(x_0^i).$$

Substituting $x_0^1 = 0$ and $x_0^2 = 1$:

$$\phi^*(x_t, t) = \frac{1}{\hat{q}_t(x_t)} \left( 0 \cdot \mathcal{N}(x_t; 0, \sigma_t^2) \cdot 0.5 + 1 \cdot \mathcal{N}(x_t; \alpha_t, \sigma_t^2) \cdot 0.5 \right),$$

$$= \frac{0.5 \cdot \mathcal{N}(x_t; \alpha_t, \sigma_t^2)}{\hat{q}_t(x_t)} = \frac{\mathcal{N}(x_t; \alpha_t, \sigma_t^2)}{\mathcal{N}(x_t; 0, \sigma_t^2) + \mathcal{N}(x_t; \alpha_t, \sigma_t^2)}.$$

Now, compute $\phi^*(g \circ x_t, t)$ by applying $g \circ x_t = x_t + 1$:

$$\phi^*(g \circ x_t, t) = \frac{\mathcal{N}(x_t + 1; \alpha_t, \sigma_t^2)}{\mathcal{N}(x_t + 1; 0, \sigma_t^2) + \mathcal{N}(x_t + 1; \alpha_t, \sigma_t^2)} < 1.$$

Next, compute $g \circ \phi^*(x_t, t)$:

$$g \circ \phi^*(x_t, t) = \frac{\mathcal{N}(x_t; \alpha_t, \sigma_t^2)}{\mathcal{N}(x_t; 0, \sigma_t^2) + \mathcal{N}(x_t; \alpha_t, \sigma_t^2)} + 1 > 1.$$

Thus,

$$\phi^*(g \circ x_t, t) < g \circ \phi^*(x_t, t),$$

since $\phi^*(g \circ x_t, t) < 1$ and $g \circ \phi^*(x_t, t) > 1$. Consequently,

$$\phi^*(g \circ x_t, t) \neq g \circ \phi^*(x_t, t).$$

This completes the counterexample, showing that $\phi^*(x_t, t)$ is not necessarily equivariant. $\qquad\square$

## C   Experimental Details

### C.1   Molecular Conformer Generation (MCG) with Flow Matching

Molecular conformer generation is a fundamental task in computational chemistry and drug discovery, where the goal is to generate plausible 3D structures (conformers) that correspond to a 2D molecular graph. Molecular conformer generation is essential for drug discovery and molecular property prediction, as the 3D structure greatly influences chemical behavior and interactions (Liu et al., 2023; Axelrod & Gómez-Bombarelli, 2020).

To model this task effectively, it is crucial to respect the underlying symmetries of molecular structures. Since conformers are invariant under global rotations and translations, one might consider the full Euclidean group $\mathsf{SE}(3)$. However, in practice, molecular structures are typically zero-centered, effectively removing the need to model translation invariance. As a result, it suffices to consider equivariance under the rotation group $\mathsf{SO}(3)$. In addition, the molecular graph may exhibit symmetry under automorphisms—permutations of atoms that preserve the graph structure—making it important to account for graph isomorphism to avoid redundant representations and ensure physically meaningful predictions.

### C.1.1 MCG - Dataset

We evaluate our method on the GEOM-QM9 dataset (Axelrod & Gomez-Bombarelli, 2022), a widely used subset of GEOM containing molecules with an average of 11 atoms. We follow the same train/validation/test split as in (Ganea et al., 2021; Jing et al., 2022), consisting of 106,586 / 13,323 / 1,000 molecules, respectively.

### C.1.2 MCG - Baselines

We compare against strong recent baselines with publicly available code, including GEODIFF (Xu et al., 2022b), GEOMOL (Ganea et al., 2021), Torsional Diffusion (Jing et al., 2022), MCF (Wang et al., 2024b), and ETFLOW (Hassan et al., 2024). GEODIFF generates structures using a roto-translationally invariant diffusion process, starting from an invariant initial density and evolving through a Markov kernel that preserves this invariance. GEOMOL predicts 3D structures by modeling torsion angles conditioned on a molecular graph, offering fast inference and good geometric validity. Torsional Diffusion generates conformers using a diffusion process in torsion space. MCF directly models 3D coordinates using a diffusion model without enforcing equivariance, relying instead on model scale to achieve strong performance. ETFLOW, the strongest of these baselines, is an equivariant flow matching model that uses a harmonic prior to encourage spatial proximity of bonded atoms. We integrate Orbit Diffusion into ETFLOW to build on its strong geometric foundation.

### C.1.3 MCG - ETFLOW with Orbit Diffusion

While Orbit Diffusion is framed within the context of diffusion models, it naturally extends to flow matching. We introduce this extension through the design of ETFLOW, which employs a harmonic prior and a flexible coupling to the data distribution. Unlike diffusion models, which fix the prior and reverse-time coupling, flow matching allows arbitrary choices for both (Tong et al., 2024), offering greater flexibility in the generative process.

Assume a coupling $q(x_0, x_1)$ between the base distribution $q_0$ and the data distribution $q_1$. For each pair $(x_0, x_1)$, define the linear interpolation:

$$I_t(x_0, x_1) = (1-t)x_0 + tx_1, \quad t \in [0, 1].$$

**Note:** In contrast to diffusion models (where $t = 0$ corresponds to data and $t = 1$ to noise), flow matching treats $x_0 \sim q_0$ as the prior sample and $x_1 \sim q_1$ as the data sample.

ETFLOW defines the conditional distribution:

$$q_t(x_t \mid x_0, x_1) = \mathcal{N}\left(x_t \mid I_t(x_0, x_1), \sigma^2 t(1-t)\right),$$

with small $\sigma$, inducing the following velocity field:

$$v_t(x_t) = x_1 - x_0 + \frac{1-2t}{2\sqrt{t(1-t)}}\epsilon, \quad \epsilon \sim \mathcal{N}(0, I).$$

Given the sampling equation:

$$x_t = (1-t)x_0 + tx_1 + \sigma\sqrt{t(1-t)}\epsilon,$$

we can express $\epsilon$ as:

$$\epsilon = \frac{x_t - (1-t)x_0 - tx_1}{\sigma\sqrt{t(1-t)}}.$$

Substituting this into $v_t(x_t)$ yields:

$$v_t(x_t) = x_1 - x_0 + \frac{1-2t}{2\sigma t(1-t)}\left(x_t - (1-t)x_0 - tx_1\right)$$

$$= \frac{1-2t}{2\sigma t(1-t)}x_t - \left(1 + \frac{1-2t}{2\sigma t}\right)x_0 + \left(1 - \frac{1-2t}{2\sigma(1-t)}\right)x_1$$

$$= h(t)x_t - g(t)x_0 + f(t)x_1.$$

The model is trained to match this target velocity using the loss:

$$\mathcal{L}_t(\phi) = \mathbb{E}_{(x_0,x_1)} \mathbb{E}_{x_t \sim q_t(\cdot|x_0,x_1)} \left[ \|\phi(x_t,t) - v_t(x_t)\|^2 \right],$$

with gradient:

$$\nabla_\phi \mathcal{L}(\phi) = \mathbb{E}_{(x_0,x_1)} \mathbb{E}_{x_t \sim q_t(\cdot|x_0,x_1)} \left[ 2(\phi(x_t,t) - v_t(x_t)) \right].$$

A Rao-Blackwellized gradient can be derived by conditioning on $x_t$:

$$\nabla_\phi \mathcal{L}(\phi) = \mathbb{E}_{x_t} \left[ 2 \left( \phi(x_t,t) - \mathbb{E}_{(x_0,x_1)|x_t} \left[ h(t)x_t - g(t)x_0 + f(t)x_1 \right] \right) \right]$$
$$= \mathbb{E}_{x_t} \left[ 2 \left( \phi(x_t,t) - h(t)x_t + g(t)\mathbb{E}_{x_0|x_t}[x_0] - f(t)\mathbb{E}_{x_1|x_t}[x_1] \right) \right]$$

We use a single sample $x_0$ to estimate $\mathbb{E}[x_0 \mid x_t]$ then use the same estimation technique as in Orbit Diffusion to compute $\mathbb{E}[x_1 \mid x_t]$, enabling efficient Rao-Blackwellized gradient estimation.

### C.1.4  MCG - Training Protocol

Instead of training from scratch, we finetune ETFlow using their public checkpoint. During training, we explicitly incorporate both forms of symmetry relevant to molecular data: discrete graph automorphisms and continuous spatial rotations. For each molecule, we uniformly sample 50 elements from its automorphism group using the `pynauty` library, capturing permutation symmetries in the 2D molecular graph structure. Simultaneously, we sample 200 elements from the rotation group $SO(3)$ to account for the continuous rotational symmetries of its 3D conformation. This symmetry-aware augmentation is applied consistently across the dataset to ensure that the model learns to respect and exploit both types of equivariances. All other training settings, including optimizer configurations and learning rate schedules, follow the defaults of ETFLOW.

### C.1.5  MCG - Evaluation Protocol

In the test set, for each molecule with $L$ ground-truth conformers, we generate $K = 2L$ conformers and evaluate their quality using standard metrics.

**Evaluation Metrics.** As a conformer $C$ represents an assignment of each atom in the molecular graph to a point in 3D space, it can be viewed as a set of vectors in $\mathbb{R}^{3n}$. To evaluate molecular conformer generation, previous works have employed two key metrics: Average Minimum RMSD (AMR) and Coverage (COV) for both Precision (P) and Recall (R). Given a molecular graph, we generate twice as many conformers as those provided by CREST. Let:

- $\{C_l^*\}_{l=1}^L$ be the set of grounth-truth conformers provided by CREST.
- $\{C_k^*\}_{k=1}^K$ be the set of generated conformers, where $K = 2L$.
- $\delta$ be a predefined RMSD threshold for considering a conformer match.

**COV-P**: Measures the proportion of generated conformers that closely match at least one ground-truth conformer.

$$\text{COV-P} = \frac{1}{K} \left| \{ k \in [1,K] \mid \exists l \in [1,L], \text{RMSD}(C_k, C_l^*) < \delta \} \right|$$

**AMR-P**: Computes the average of the minimum RMSD values between each generated conformer and its closest ground-truth conformer.

$$\text{AMR-P} = \frac{1}{K} \sum_{k=1}^K \min_{l=1}^L \text{RMSD}(C_k, C_l^*)$$

**COV-R**: Measures the proportion of ground-truth conformers that have at least one close-enough generated conformer.

$$\text{COV-R} = \frac{1}{L} \left| \{ l \in [1,L] \mid \exists k \in [1,K], \text{RMSD}(C_k, C_l^*) < \delta \} \right|$$

**AMR-R**: Computes the average of the minimum RMSD values between each ground-truth conformer and its closest generated conformer.

$$\text{AMR-R} = \frac{1}{L} \sum_{k=1}^L \min_{l=1}^K \text{RMSD}(C_k, C_l^*)$$

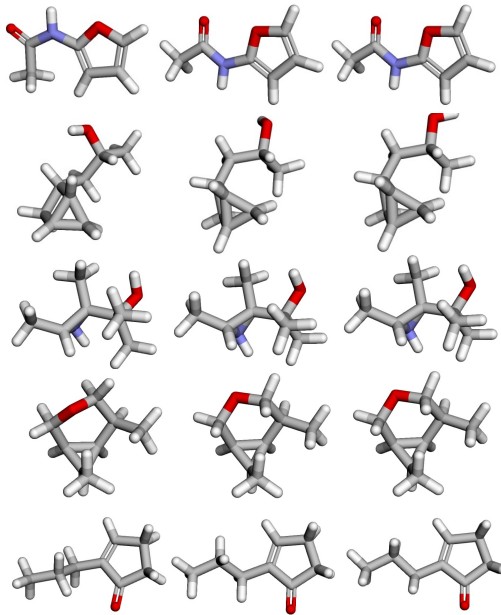

Figure 7: Molecular conformers generated by ETFLOW (left), + [OrbDiff] (center), and ground-truth (right).

### C.1.6 MCG - Full results

Table 6: Molecular conformer generation performance on GEOM-QM9. [*] Reported in the original paper. [†] Obtained using the published checkpoint. [‡] From our reimplementation trained from scratch.

| Models | Recall | | | | | | Precision | | | | | |
|---|---|---|---|---|---|---|---|---|---|---|---|---|
| | Cov@0.1 (↑) | | Cov@0.5 (↑) | | AMR (↓) | | Cov@0.1 (↑) | | Cov@0.5 (↑) | | AMR (↓) | |
| | Mean | Median | Mean | Median | Mean | Median | Mean | Median | Mean | Median | Mean | Median |
| GEODIFF | - | - | 76.5 | **100.0** | 0.297 | 0.229 | - | - | 50.0 | 33.5 | 1.524 | 0.510 |
| GEOMOL[†] | 28.4 | 0.0 | 91.1 | **100.0** | 0.224 | 0.194 | 20.7 | 0.0 | 85.8 | **100.0** | 0.271 | 0.243 |
| Torsional Diff.[†] | 37.7 | 25.0 | 88.4 | **100.0** | 0.178 | 0.147 | 27.6 | 12.5 | 84.5 | **100.0** | 0.221 | 0.195 |
| MCF[†] | 81.9 | **100.0** | 94.9 | **100.0** | 0.103 | 0.049 | 78.6 | **93.8** | 93.9 | **100.0** | **0.113** | 0.055 |
| ETFLOW[*] | - | - | 96.5 | **100.0** | 0.073 | 0.047 | - | - | 94.1 | **100.0** | 0.098 | 0.039 |
| ETFLOW[†] | 79.5 | **100.0** | 93.8 | **100.0** | 0.096 | 0.037 | 74.4 | 83.3 | 88.7 | **100.0** | 0.142 | 0.066 |
| ETFLOW[‡] | 81.4 | **100.0** | 94.4 | **100.0** | 0.092 | 0.039 | 74.6 | 85.5 | 89.1 | **100.0** | 0.145 | 0.064 |
| + [OrbDiff] | **85.4** | **100.0** | **96.3** | **100.0** | **0.074** | **0.027** | **80.2** | **93.9** | 91.9 | **100.0** | **0.113** | **0.042** |

### C.2 Crystal Structure Prediction (CSP)

Crystal Structure Prediction (CSP) is the process of identifying the most stable three-dimensional arrangement of atoms in a crystalline solid, given only its chemical formula. This task lies at the heart of computational materials science, as the resulting crystal structure dictates key physical and chemical properties—including thermodynamic stability, electronic conductivity, and chemical reactivity (Wei et al., 2024; Shao et al., 2021; Kim et al., 2025). Despite its importance, CSP remains a formidable challenge due to the immense combinatorial search space of atomic positions and the presence of complex symmetry constraints that define equivalent configurations (Oganov & Glass, 2006). Efficiently navigating this landscape to discover low-energy, physically plausible structures continues to be a central focus of the field.

We also evaluate our method on a related task introduced by TGDMat (DAS et al., 2025), where crystal structures are generated based on textual descriptions of the desired materials. This setting reflects a more user-centric interface for materials design, where scientists or domain experts can

specify target properties or structural features in natural language. The task includes two levels of textual conditioning: long descriptions, which provide detailed structural and compositional information, and short descriptions, which are more concise and easier to obtain. Supporting text-to-structure generation enables more accessible and flexible workflows in materials discovery, especially in scenarios where precise structural data may be unavailable.

### C.2.1   CSP - Dataset

We evaluate our method on two used CSP benchmarks: Perov-5 (Castelli et al., 2012a,b) and MP-20 (Jain et al., 2013). These datasets encompass a broad range of inorganic crystal compositions. Perov-5 comprises 18,928 perovskite structures, which share a common structural motif but vary in elemental composition. In contrast, MP-20 includes 45,231 stable inorganic materials curated from the Materials Project database (Jain et al., NA), offering a more diverse set of crystal systems and chemistries.

### C.2.2   CSP - Baselines

For comparison, we use the three strongest baselines from the DiffCSP paper (Jiao et al., 2023): P-cG-SchNet (Gebauer et al., 2022), CDVAE (Xie et al., 2022), and DiffCSP, all with publicly available implementations.

### C.2.3   CSP — DiffCSP and TGDMat with Orbit Diffusion

DiffCSP (Jiao et al., 2023) is a diffusion-based framework for crystal structure prediction that jointly models lattice parameters and atomic positions while respecting the fundamental symmetries of crystalline materials. Since TGDMat is built upon DiffCSP, we focus on describing how Orbit Diffusion integrates with DiffCSP; the same integration applies directly to TGDMat.

DiffCSP formulates the task as a *joint diffusion process* with two interconnected components: one for the *lattice* and one for the *atomic coordinates*. The lattice defines the shape and scale of the unit cell, while the coordinates specify atomic positions in fractional units relative to the lattice vectors. To capture the relevant symmetries, the lattice diffusion is $O(3)$-equivariant (invariant under rotations and reflections), and the coordinate diffusion is both *permutation equivariant* and *periodic translation equivariant*, reflecting atomic indistinguishability and lattice periodicity.

Since the lattice involves only a few parameters, it is a relatively simple subtask. We therefore concentrate on the more challenging component: generating atomic coordinates. To ensure periodic translation equivariance, DiffCSP defines a forward diffusion process based on the kernel $\hat{q}^G(x_t \mid x_0)$, modeled as a *Wrapped Normal* distribution—a periodic analogue of the Gaussian—ensuring the diffusion respects the toroidal geometry of fractional coordinates. Specifically,

$$\hat{q}^G(x_t \mid x_0) \propto \sum_{z \in \mathbb{Z}^d} \exp\left(-\frac{\|x_t - x_0 + z\|^2}{2\sigma_t^2}\right), \tag{31}$$

which defines a valid density on the torus $\mathbb{T}^d$. One can verify that $\hat{q}^G(g \circ x_t \mid g \circ x_0) = \hat{q}^G(x_t \mid x_0)$ for any $g$ in the periodic translation group (Jiao et al., 2023), confirming its equivariance. Consequently, all our theoretical results (e.g., Theorem 1 and Theorem 2) hold when applying Orbit Diffusion to DiffCSP under the periodic translation group.

**Orbit Diffusion with non-uniform group sampling.**   Rather than sampling uniformly from the group $G$, for the periodic translation group we propose sampling translation elements from a Wrapped Normal distribution. We refer to this variant as `OrbDiff_WN`. Formally,

$$\nu_t(g) \propto \sum_{z_g \in \mathbb{Z}^{d_g}} \exp\left(-\frac{\|m_g + z_g\|^2}{2\sigma_g(t)^2}\right), \tag{32}$$

where $m_g$ is the translation vector corresponding to group element $g$, and $\sigma_g(t)$ is the time-dependent bandwidth. In our experiments with both DiffCSP and TGDMat, we set $\sigma_g(t) = 2\sigma_t$. To sample from this distribution, we first draw $\epsilon \sim \mathcal{N}(0, I)$ in $\mathbb{R}^3$, and then compute $m_g = \sigma_g(t)\epsilon \mod 1$. We sample 1000 such group elements per step to form the group approximation.

**More training details.** All models were trained on a single NVIDIA GeForce RTX 4090 GPU. TGDMat was trained for 1,500 epochs, while DiffCSP was trained for 500 epochs. On the MP20 dataset, each epoch took roughly 15 seconds, resulting in total training times of 6.25 hours for TGDMat and 2.08 hours for DiffCSP. On the Perov-5 dataset, each epoch took about 5 seconds, corresponding to 2.08 hours (TGDMat) and 0.69 hours (DiffCSP) of training time.

### C.2.4 CSP - Evaluation Protocol

To evaluate crystal structure prediction, we randomly generate one sample for each structure in the test set. We then calculate two metrics: the Match Rate and the average Root Mean Square Distance (RMSD) across the test set. We repeat this procedure three times and report the median values of these metrics for more reliability.

**Match rate**: The Match Rate is defined as the proportion of predicted structures that successfully match the corresponding ground-truth structures in the test set. Specifically, it is calculated as follows:

$$\text{Match Rate} = \frac{\text{Number of matched structure pairs}}{\text{Total number of test samples}}$$

Following previous works (Jiao et al., 2023; Xie et al., 2022), we use the `StructureMatcher` class from the `pymatgen` library to determine structure matching. The matching process is based on the following criteria:

- Length Tolerance (*ltol*): 0.5 (fractional length tolerance).

- Site Tolerance (*stol*): 0.3 (fraction of the average free length per atom)

- Angle Tolerance (*atol*): 10 (in degrees)

The `StructureMatcher` algorithm aligns the lattice vectors of two structures. If the tolerance criteria are satisfied, the structures are considered matched.

**RMSD**: For an alignment between lattices of two structures, `StructureMatcher` continues to align atoms to compute the average RMSD. The process is repeated for all possible lattices to find the smallest RMSD. Then the Average RMSD is computed as the average of the smallest RMSD of all matched structure pairs.

$$\text{RMSD} = \frac{1}{N_{\text{matched}}} \sum_{i=1}^{N_{\text{matched}}} \text{RMSD}(\text{generated}_i, \text{ground-truth}_i)$$

Here, $N_{\text{matched}}$ is the total number of matched structures. Unmatched structures are excluded from the calculation.

Ideally, we aim for a high Match Rate and a low RMSD. A low Match Rate with a low RMSD is not useful because unmatched samples are effectively treated as having very high RMSD. Thus, RMSD alone cannot fully capture prediction quality. We emphasize that Match Rate is more critical, especially during initial screening, where we prioritize valid structures over perfectly matched ones.

### C.2.5 CSP - Full Results and Visualizations

For the TGDMat models (TGDMat (S) and TGDMat (L)), we trained both from scratch, experimenting with and without the proposed loss functions: `[OrbDiff_U]` and `[OrbDiff_WN]`. Meanwhile, the DiffCSP model was trained using our proposed losses, while the baseline models (P-cG-SchNet, CDVAE, and DiffCSP) rely on the results reported in the original DiffCSP paper. Quantitative results are summarized in Tables 7 and 8 with qualitative comparisons shown in Figures 8 and 9.

Table 7: Text-guided CSP with TGDMat.

| Method | Perov-5 | | MP-20 | |
|---|---|---|---|---|
| | Match (↑) | RMSE (↓) | Match (↑) | RMSE (↓) |
| TGDMat (S) | 59.39 | 0.066 | 59.90 | 0.078 |
| + [OrbDiff_U] | 63.51 | 0.062 | 56.50 | 0.085 |
| + [OrbDiff_WN] | **65.57** | **0.054** | **61.29** | **0.072** |
| TGDMat (L) | 95.17 | 0.013 | 61.91 | 0.081 |
| + [OrbDiff_U] | 95.88 | **0.012** | 65.94 | **0.069** |
| + [OrbDiff_WN] | **95.98** | **0.012** | **66.74** | **0.069** |

Table 8: Crystal Structure Prediction (CSP).

| Method | Perov-5 | | MP-20 | |
|---|---|---|---|---|
| | Match (↑) | RMSE (↓) | Match (↑) | RMSE (↓) |
| P-cG-SchNet | 48.22 | 0.418 | 15.39 | 0.376 |
| CDVAE | 45.31 | 0.114 | 33.90 | 0.105 |
| DiffCSP | 52.02 | 0.076 | 51.49 | 0.063 |
| + [OrbDiff_U] | 52.29 | 0.078 | 54.47 | 0.054 |
| + [OrbDiff_WN] | **52.39** | **0.069** | **55.70** | **0.053** |

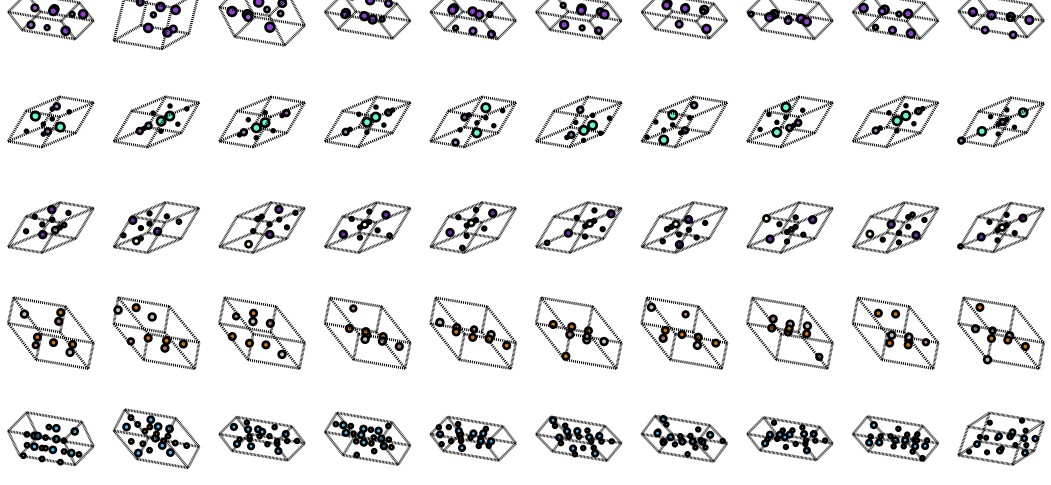

DiffCSP  + [OrbDiff_U] + [OrbDiff_WN] TGDMAT (S)  + [OrbDiff_U] + [OrbDiff_WN] TGDMAT (L)  + [OrbDiff_U] + [OrbDiff_WN] Ground-truth

Figure 8: Qualitative comparison of Crystal Structure Predictions by 9 models, including DiffCSP, TGDMAT short and TGDMAT long with baselines, `OrbDiff_U`, and `OrbDiff_WN` against ground-truth samples on randomly selected samples from MP-20 dataset.

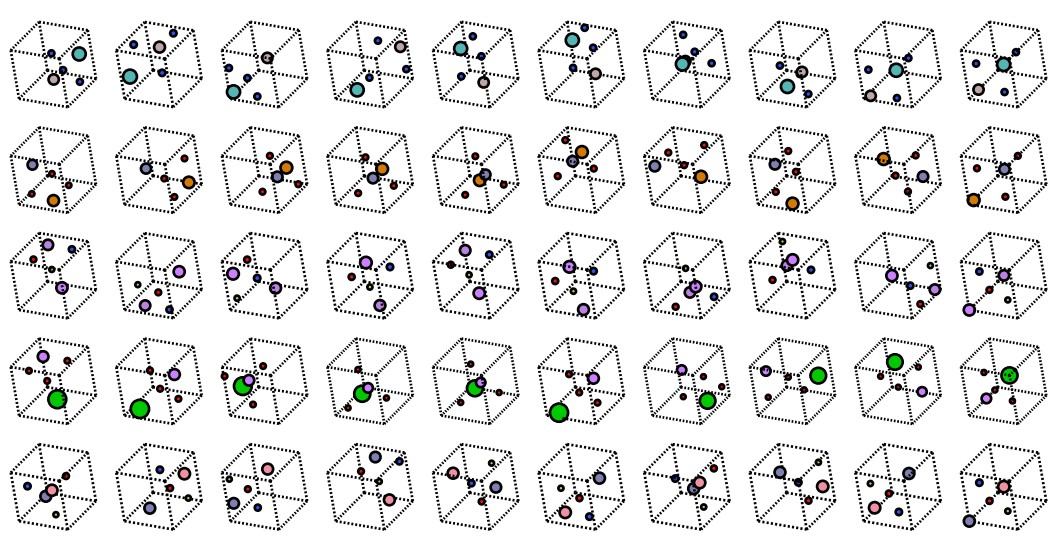

DiffCSP  + [OrbDiff_U] + [OrbDiff_WN] TGDMAT (S)  + [OrbDiff_U] + [OrbDiff_WN] TGDMAT (L)  + [OrbDiff_U] + [OrbDiff_WN] Ground-truth

Figure 9: Qualitative comparison of Crystal Structure Predictions by 9 models, including DiffCSP, TGDMat (S) and TGDmat (L) with baselines, `OrbDiff_U`, and `OrbDiff_WN` against ground-truth samples on randomly selected samples from Perov-5 dataset.

## C.3 Protein Structure Generation (PSG) with Non-Equivariant Denoiser PROTEINA

Protein structure generation focuses on sampling physically valid 3D conformations of proteins by learning a probabilistic distribution over either atomistic or coarse-grained representations. This generative task plays a crucial role in de novo protein design and has broad implications for understanding protein folding and function (Jing et al., 2023; Wu et al., 2022; Watson et al., 2023; Huguet et al., 2024; Jing et al., 2023). Unlike traditional structure prediction, which aims to infer a single or most likely conformation, generative models must capture the full structural manifold—accounting for the inherent $SO(3)$ rotational symmetry of protein backbones, maintaining biochemical realism, and respecting physical constraints such as bond lengths and steric clashes (Gaujac et al., 2024; Geffner et al., 2025). Effectively modeling these aspects ensures that generated structures are not only diverse but also biologically and physically plausible.

### C.3.1 PSG - Baselines

We list the baselines used by the work of (Geffner et al., 2025) as follows:

- FrameDiff (Yim et al., 2023b)
- FoldFlow (Bose et al., 2024)
- FrameFlow (Yim et al., 2023a)
- ESM3 (Hayes et al., 2025)
- Chroma (Ingraham et al., 2023)
- RFDiffusion (Watson et al., 2023)
- Proteus (Wang et al., 2024a)
- Genie2 (Lin et al., 2024)

**PROTEINA.** PROTEINA is a large-scale, flow-based model for generating protein backbones, built on a scalable transformer architecture. Unlike equivariant models, it does not enforce equivariance, granting more architectural flexibility. This choice allows the use of powerful transformer networks with hundreds of millions of parameters, enabling PROTEINA to effectively learn from large datasets. As a result, it excels at modeling complex protein structures by balancing expressiveness and computational scalability.

There are three variants of PROTEINA:
(i) $\mathcal{M}_{\text{FS}}$, a 200M-parameter transformer with an additional 15M parameters in triangle layers, trained on the **Foldseek AFDB clusters** dataset $\mathcal{D}_{\text{FS}}$, which includes 555,318 structures of lengths 32–256;
(ii) $\mathcal{M}_{\text{FS}}^{\text{no-tri}}$, a simplified version without triangle layers, also with 200M parameters and trained on the same dataset;
(iii) $\mathcal{M}_{21M}$, a 400M-parameter transformer with 15M triangle parameters, trained on a **high-quality filtered AFDB subset** $\mathcal{D}_{21M}$, comprising approximately 21M structures. $\mathcal{M}_{21M}$ represents the current state of the art in designability modeling.

### C.3.2 PROTEINA with Orbit Diffusion

We apply Orbit Diffusion to the simplest variant of PROTEINA, namely $\mathcal{M}_{\text{FS}}^{\text{no-tri}}$, by fine-tuning the public checkpoint released by Geffner et al. (2025), since we do not have access to the extensive compute resources used for the original training. Our fine-tuning setup uses 4 A100 GPUs for 24 hours on the same dataset, whereas the original training employed 96 GPUs.

The key symmetry group in protein structure generation is $SO(3)$, which represents 3D rotations. To exploit this symmetry using `OrbDiff`, we apply 10,000 uniformly sampled random rotations to each training sample as part of our Rao-Blackwell estimator, enhancing both the efficiency and stability of the flow matching process.

The only change we introduce to the original model is replacing the conditional flow matching loss with our proposed flow matching objective (see Appendix C.1.3).

For a fair comparison, we also fine-tune $\mathcal{M}_{\text{FS}}^{\text{no-tri}}$ (+ [finetune]) using the original loss under the same computational budget, with a batch size of 8 and 32 gradient accumulation steps.

### C.3.3 PSG - Evaluation Protocol

To evaluate the quality of our generated protein backbones, we rely on three widely used metrics: designability, diversity, and novelty. Following the protocol established by Geffner et al. (2025), we generate 500 samples total—100 for each length in {50, 100, 150, 200, 250}—and compute all metrics on this dataset. Among these, **designability is the most critical metric**, as it directly reflects the biological feasibility of the generated structures and serves as the foundation for the other two metrics.

**Designability.** Designability measures whether a backbone structure can realistically be encoded by an amino acid sequence. For each generated backbone, we produce 8 candidate sequences using ProteinMPNN (Dauparas et al., 2022) with a sampling temperature of 0.1. These sequences are then folded using ESMFold (Lin et al., 2023), and the root mean square deviation (RMSD) is calculated between each predicted structure and the original backbone. A backbone is deemed designable if at least one sequence folds with an RMSD below 2Å, where this minimum RMSD is known as the self-consistency RMSD (scRMSD).

Since **diversity and novelty are computed only on designable samples**, accurate assessment of designability is essential for interpreting the other metrics meaningfully.

The designability score for a model is reported as the fraction of samples deemed designable. Additionally, we report the average scRMSD across all samples, allowing for a more nuanced comparison between our model and existing baselines.

**Diversity.** We evaluate diversity among the designable samples using two approaches. First, we compute the average pairwise TM-score within each length group (50, 100, 150, 200, and 250) as a measure of structural variation. Lower average TM-scores indicate greater diversity.

Second, we calculate diversity (cluster) by grouping designable samples into clusters based on a TM-score threshold of 0.5. Each cluster contains samples with pairwise TM-scores above this threshold. Diversity (cluster) is then defined as the ratio of the total number of clusters to the number of designable samples. A higher ratio reflects a larger number of distinct structural groups relative to the sample size, signaling increased diversity.

**Novelty.** Novelty measures how structurally distinct the designable samples are compared to known protein structures. For each designable backbone, we compute the TM-score to its closest match in two reference sets: the Protein Data Bank (PDB) and the $\mathcal{D}_{FS}$ dataset used for training. The average of these maximum TM-scores across all designable samples is reported as the novelty score. Lower values indicate the model generates structures that are more novel relative to both established experimental data and the training distribution.

### C.3.4 PSG - Full results

Table 9: **Protein Structure Generation.** Full comparison with baseline models; all baseline results are taken from (Geffner et al., 2025). "+ [finetune]" indicates $\mathcal{M}_{FS}^{\text{no-tri}}$ finetuned with the original loss, while "+ [OrbDiff]" denotes finetuning with OrbDiff.

| Model | Designability | | Diversity | | Novelty | |
|---|---|---|---|---|---|---|
| | Fraction ($\uparrow$) | scRMSD ($\downarrow$) | Cluster ($\uparrow$) | TM-score ($\downarrow$) | PDB ($\downarrow$) | AFDB ($\downarrow$) |
| FrameDiff | 65.4 | - | 0.39 | 0.40 | 0.73 | 0.75 |
| FoldFlow (base) | 96.6 | - | 0.42 | 0.75 | 0.75 | 0.77 |
| FoldFlow (stoc.) | 97.0 | - | 0.61 | 0.38 | 0.62 | 0.68 |
| FoldFlow (OT) | 97.2 | - | 0.37 | 0.41 | 0.71 | 0.75 |
| FrameFlow | 88.6 | - | 0.59 | 0.34 | 0.79 | 0.80 |
| ESM3 | 22.0 | - | 0.52 | 0.57 | 0.70 | 0.75 |
| Chroma | 78.8 | - | 0.42 | 0.43 | 0.77 | 0.76 |
| RFDiffusion | 94.4 | - | 0.46 | 0.34 | 0.79 | 0.80 |
| Proteus | 94.4 | - | 0.42 | 0.43 | 0.77 | 0.80 |
| Genie2 | 95.2 | - | 0.59 | 0.38 | **0.63** | 0.69 |
| $\mathcal{M}_{21M}$ | **99.0** | **0.72** | 0.30 | 0.39 | 0.81 | 0.84 |
| $\mathcal{M}_{FS}^{\text{no-tri}}$ | 93.8 | 1.04 | **0.62** | **0.36** | **0.69** | **0.76** |
| + [finetune] | 93.8 | 1.00 | 0.54 | 0.37 | 0.74 | 0.83 |
| + [OrbDiff] | **95.6** | **0.93** | 0.52 | 0.37 | 0.74 | 0.83 |

The state-of-the-art model $\mathcal{M}_{21M}$, with 400M parameters and trained on a large, high-quality dataset, achieves the highest designability (99.0%) and lowest scRMSD (0.72), reflecting its strong reconstruction capability. However, this comes at the expense of diversity and novelty: it exhibits the lowest diversity score (Cluster: 0.30) and higher novelty metrics (PDB: 0.81, AFDB: 0.84), indicating reduced structural variety and generalization.

In comparison, our base model $\mathcal{M}_{FS}^{no\text{-}tri}$ already achieves competitive performance (designability: 93.8%, Cluster: 0.62, PDB: 0.69), and naive finetuning with data augmentation ("+ [finetune]") fails to improve designability or diversity. Notably, our method ("+ [OrbDiff]") improves designability to 95.6% and reduces scRMSD to 0.93, while preserving competitive diversity (Cluster: 0.52) and novelty (PDB: 0.74). This highlights that OrbDiff is an effective finetuning strategy to enhance functional accuracy without fully sacrificing structural diversity—achieving a better trade-off than both naive finetuning and heavily overparameterized models.

