# OpenReview forum: "Rao-Blackwell Gradient Estimators for Equivariant Denoising Diffusion"
_NeurIPS.cc/2025/Conference — NeurIPS 2025 poster_

### Official Review · Reviewer_veFd · 2025-06-23

**Clarity:** 3
**Significance:** 3
**Originality:** 3
**Rating:** 4
**Confidence:** 4

**Summary:**

This paper introduces OrbDiff, a framework for training group-invariant generative diffusion models that leverages Rao–Blackwellized gradient estimators to reduce variance. The method can be used in both equivariant model designs and data augmentation schemes and provably improves the variance of standard gradient estimators. OrbDiff is tested on molecular conformation generation, crystal structure prediction, and protein design. Experiments suggest improvements in gradient variance, and convergence speed.

**Questions:**

1. ___Clarification on the Data Assumptions:___ The paper uses symmetrized distributions $\hat{q}^G$ to define the loss under data augmentation. Is the assumption that the underlying data distribution $q$ is already $G$-invariant? If so, symmetrization doesn't do anything. If not, what justifies learning a $G$-invariant distribution in the first place? A more formal treatment of this modeling assumption would clarify the motivation and benefit the paper. E.g., I'm assuming that the "true" data distribution is $G$-invariant but the samples a biased in some way.

2. ___Justification of the High Variance Claim:___ The paper states that the standard gradient estimator (Equations (4) and (5)) suffers from high variance. However, this estimator is just the typical diffusion gradient with augmented samples. The claim that it has higher variance than in the non-invariant case is plausible but not theoretically motivated explicitly. Could the authors provide either a mathematical argument or additional experiments to support this claim? This would significantly improve the argument of the of paper.

3. ___Clarity of Theorem 1:___ The statement of Theorem 1 compares the variance of two gradient estimators, but Equations (5) and (7) define expectations, not estimators. To analyze variance, one needs to specify how the estimators are constructed—e.g., which quantities are sampled via Monte Carlo. The theorem is likely correct with a reasonable interpretation, but a clearer statement would benefit the paper.

4. ___Notes:___
    - The method is referred to as “Rao–Blackwell Gradient Estimation” but the Rao–Blackwell theorem is never explicitly mentioned in the main body (only in the proof of Theorem 1 in the Appendix). It would be helpful to summarize the relevant argument in the main text rather than relegating it to the Appendix.
    - Consider presenting the RB loss $\mathcal{L}_t^\mathrm{RB}(\phi)$ (Equation (13)) earlier in the paper, as it is more interpretable and makes the goal of the method clearer. Starting from the loss makes it easier to understand what the method is doing and how it works in practice.

**Ethical Concerns:**

["NO or VERY MINOR ethics concerns only"]

**Final Justification:**

The paper presents a clean, general method for reducing gradient variance in symmetry-aware diffusion models, with solid theory and strong results across tasks. While the main text underexplains some theoretical points, the rebuttal clarifies them well. My main concern is impact: the paper doesn't clearly show that high variance is (a) especially severe in equivariant models or (b) meaningfully harmful. Overall, this is a well-motivated contribution, but the paper would benefit from earlier clarity and stronger evidence that the addressed problem is significant.

**Limitations:**

yes

**Paper Formatting Concerns:**

No major formatting concerns.

**Quality:**

3

**Strengths And Weaknesses:**

**Strengths:**
- Conceptually simple method with provable variance reduction guarantees.
- The method is broadly applicable in symmetry-aware generative modeling. It works with both equivariant architectures and data augmentations.
- Extensive empirical evaluation across diverse domains.

**Weaknesses:**
- The motivation for focusing on the variance in the gradient estimator is not clearly formalized or empirically quantified outside the synthetic setup.
- The connection to the Rao–Blackwell theorem is not clearly explained in the main text, despite being central to the method.

---

> ### Author Rebuttal · Authors · 2025-07-29
>
> Thank you for your valuable feedback and for pointing out the areas of improvement in our manuscript.
>
>
> ### **Weakness #1: The motivation for focusing on the variance in the gradient estimator is not clearly formalized or empirically quantified outside the synthetic setup**
>
> We appreciate the reviewer’s comment regarding the motivation behind our focus on the variance in the gradient estimator. Our primary motivation is based on the insight that high-variance gradient estimators can lead to unstable or inefficient optimization. Moreover, prior work has shown that reducing gradient variance during the training of diffusion models can lead to improved final performance [1, 2, 3].
>
> In revisiting our analysis during the rebuttal—prompted particularly by Reviewer 2’s question on the potential bias of OrbDiff—we identified an important theoretical property that was not highlighted in the original manuscript: **the OrbDiff gradient estimator is unbiased**, even though it relies on a proposal distribution that lead to a biased estimation of $\mathbb{E}[x_0 \mid x_t]$. This finding further supports our motivation for using OrbDiff, as it enables variance reduction without introducing bias into the gradient estimate.
>
> Additionally, OrbDiff incurs only minimal extra computational and memory overhead. For the full proof of the unbiasedness result, please refer to **our response to Reviewer 1(j92t)**.
>
> [1] Variational Diffusion Models
>
> [2] Stable Target Field for Reduced Variance Score Estimation in Diffusion Models
>
> [3] Improved Denoising Diffusion Probabilistic Models
>
>
> ### **Weakness #2: The connection to the Rao–Blackwell theorem is not clearly explained in the main text, despite being central to the method**
>
> Thank you for pointing out this concern. We will include the RB theorem in the main paper to show the connection.
>
> ### **Question #1: Clarification on the Data Assumptions**
>
> We appreciate the reviewer’s insightful question. To clarify, we do assume that the underlying data-generating distribution $q$ is **G**-invariant. However, the observed distribution $\hat{q}$​ (from which we obtain training samples) is generally not **G**-invariant due to dataset biases. For example, in molecular datasets, each molecule may be saved in a canonical but arbitrary orientation, even though physically all rotated versions are equally probable under $q$.
>
> As a result, training directly on $\hat{q}​$ would lead to a model that may not respect the underlying symmetry. This is a widely recognized issue in the literature, and standard solutions include:
> - Using equivariant networks, which enforce symmetry by architectural design.
> - Using data augmentation, where samples are transformed by group actions during training.
>
> Both approaches can be seen as implicitly training on the symmetrized distribution $\hat{q}^G$ [4]. We formulate the loss under the symmetrized distribution to provide a unified framework for analyzing existing methods like data augmentation and equivariant architectures. This perspective clarifies their implicit inductive biases and reveals new opportunities for improvement—most notably, the construction of lower-variance gradient estimators, which directly motivates OrbDiff.
>
> [4] Equivariant score-based generative models provably learn distributions with symmetries efficiently
>
> ### **Question #2: Justification of the High Variance Claim**
>
> Thank you for pointing this out. We apologize for the confusion caused by our original statement. The claim in the paper that “this estimator, used in data augmentation, suffers from high variance” at line 126 in the paper is imprecise.
>
> We meant to convey that gradient estimators in diffusion model training in general, whether with or without data augmentation, are known to suffer from high variance, as supported by prior work [1, 2, 3].
>
> We will revise the statement to “Gradient estimators used in diffusion model training—including those with data augmentation—are known to suffer from high variance.”
>
> We appreciate the reviewer’s careful reading and will include additional citations and empirical evidence to clarify this point.
>
> [1] Variational Diffusion Models
>
> [2] Stable Target Field for Reduced Variance Score Estimation in Diffusion Models
>
> [3] Improved Denoising Diffusion Probabilistic Models
>
> ### **Question #3: Clarity of Theorem 1**
>
> Thank you for the helpful observation. You're right that Equations (5) and (7) define expectations, while variance applies to their finite-sample estimators. In practice, we use standard minibatch estimators during training, and Theorem 1 compares their variances through the expectations they approximate.
>
> We will clarify this in the paper and explain that the result is motivated by the Rao-Blackwell theorem: replacing a random group sample with its conditional expectation leads to a lower-variance estimator. This justifies our analysis at the level of expectations and explains the benefit of the OrbDiff estimator in practice.
>
> ### **Question #4: Suggestion for improving the paper**
>
> Thank you for the suggestion. We agree and will move the key idea behind the Rao–Blackwell theorem into the main text to better motivate our method. We’ll also present the Rao-Blackwellized loss (Eq. 13) earlier in the paper, as it provides a clearer and more intuitive starting point for understanding OrbDiff in practice.

---

> > ### Comment · Reviewer_veFd · 2025-08-03
> > **Response to Authors' Rebuttal**
> >
> > I thank the authors for their detailed response, and for addressing my questions, reaffirming my positive assessment of the paper.

---

> > > ### Author Response · Authors · 2025-08-03
> > > **Response to Reviewer's comment**
> > >
> > > Thank you for your thoughtful and positive review. We're glad our rebuttal addressed your concerns and helped clarify the contributions of the paper.
> > >
> > > Given your reaffirmed positive assessment, we would appreciate it if you could consider whether a stronger score might better reflect your view of the paper's quality and contribution.
> > >
> > > Regardless, we sincerely appreciate your time and constructive feedback.

---

### Official Review · Reviewer_LfDH · 2025-06-23

**Clarity:** 3
**Significance:** 3
**Originality:** 3
**Rating:** 5
**Confidence:** 3

**Summary:**

This paper considers a novel variance reduction technique for the gradients of the squared loss in diffusion models with symmetry through Rao-Blackwell's Theorem. It offers practical guidance called Orbit Diffusion that can be applied to either equivariant or non-equivariant neural network architecture. Numerical simulations on various datasets verify the usefulness of the proposed method. In general, this work offers a practical bias-variance trade-off dedicated to the equivariant diffusion models.

**Questions:**

1. Line 94, $q(x_T) =\mathcal{N}(0,I)$ should be $q(x_T) \approx\mathcal{N}(0,I)$ since you would not achieve stationarity in finite time. Also, the notation is confusing, I think it might be better to replace $q(x_0), q(x_T)$ by $q_0(x), q_T(x)$.

2. What is $D$ in line 123? Has it been defined before?

3. For Eq. (4) to be equivalent to Eq. (5), you may need to make certain assumptions on the noising kernel.

4. What's the computational burden for the SNIS?

5. What if you don't bias as you did in lines 170-171? How is the performance improvement?

**Ethical Concerns:**

["NO or VERY MINOR ethics concerns only"]

**Final Justification:**

I thank the authors for the detailed response and the additional experiments, both of which have addressed my questions and concerns.

**Limitations:**

Yes.

**Paper Formatting Concerns:**

None.

**Quality:**

3

**Strengths And Weaknesses:**

Strength: The paper is written clearly, and the idea is simple but interesting. It offers practical contributions to the training of equivariant diffusion models.

Weaknesses: How the variance reduction of the gradient improves performance is still underexplored. Orbit Diffusion includes bias.

---

> ### Author Rebuttal · Authors · 2025-07-29
>
> Thank you for your valuable feedback and for pointing out the areas of improvement in our manuscript.
>
> ### **Weakness #1: How the variance reduction of the gradient improves performance is still underexplored.**
>
> To study the practical benefit of variance reduction, we conducted extensive experiments across diverse tasks—molecular conformer generation, crystal structure prediction, and protein structure generation. In all these settings, we observe consistent improvements in model performance when using OrbDiff, demonstrating that lower-variance gradients translate into better downstream results. Furthermore, from diffusion research literature, high-variance gradient estimators can lead to unstable or inefficient optimization. Specifically, prior work has shown that reducing gradient variance during the training of diffusion models can lead to improved final performance [1, 2, 3].
>
> [1] Variational Diffusion Models
>
> [2] Stable Target Field for Reduced Variance Score Estimation in Diffusion Models
>
> [3] Improved Denoising Diffusion Probabilistic Models
>
> ### **Weakness #2: Orbit Diffusion includes bias**
>
> Thank you for highlighting this important point. We emphasize that the gradient estimator used to train the model **remains unbiased**, even though OrbDiff’s SNIS-based estimate of the inner expectation $\mathbb{E}[x_0 \mid x_t]$ is biased. This happens because the bias cancels out when computing the outer expectation of the gradient. This subtle but crucial distinction was not clearly conveyed in the original version of the paper. We appreciate the reviewer for prompting this clarification and will revise the text to clearly state that OrbDiff yields unbiased gradient estimates, despite relying on a biased proposal for efficiency.
>
> Please find the **proof in the response to Reviewer 1’s comment.**
>
> ### **Question #1: Notation clarity**
>
> Thank you for the helpful feedback. We agree with the first point and will revise the text to write $q(x_T) \approx \mathcal{N}(0, I)$ to reflect the fact that exact stationarity may not be achieved in finite time.
>
> Regarding the second point, we acknowledge the potential for confusion, but we follow the standard notation in the diffusion literature (e.g.,[1]), where $q(x_t)$ denotes the distribution of $x_t$​ at time $t$. We will clarify this in a footnote or early in the notation section to help readers unfamiliar with this convention.
>
> [1] Denoising Diffusion Probabilistic Models
>
> ### **Question #2: What is D in line 123?**
>
> We thank the reviewer for the question. It is the dataset. We will add the definition of the notation to the manuscript.
>
> ### **Question #3: The equivalent of equations (4) and (5)**
>
> Thank you for the thoughtful comment. We realize that the equivalence between Eqs. (4) and (5) may not be immediately clear due to the notation. In Eq. (4), we write $x_0 \sim \hat{q}^G(x_0)$, whereas in Eq. (5), the distribution of $x_t$ is written as $\hat{q}^G_t(x_t)$, which could give the impression that different distributions are being used.
>
> To clarify: both equations are expressing expectations under the same joint distribution over $(x_0,x_t)$, where $x_0 \sim \hat{q}^G(x_0)$, and $x_t \sim \hat{q}^G_t(x_t \mid x_0)$, defined via a group-equivariant noising process. Since this defines a well-formed joint distribution $\hat{q}^G_t(x_0, x_t)$, we can apply the law of iterated expectations in either order, giving equivalent forms: $\mathbb{E} _{x_0 \sim \hat{q}^G(x_0)}\mathbb{E} _{x_t \sim \hat{q}^G_t(x_t \mid x_0)}[⋅]  = \mathbb{E} _{x_t \sim \hat{q}^G_t(x_t)}\mathbb{E} _{x_0 \sim \hat{q}^G_t(x_0 \mid x_t)}[⋅]$
>
> We will update the text to clarify this point explicitly and refine the notation to avoid potential confusion between $\hat{q}^G(x_0)$ and $\hat{q}^G_0(x_0)$, which we used interchangeably.
>
> ### **Question #4: Computational burden of SNIS**
>
> Thank you for the thoughtful question. It's worth highlighting that the main cost during training typically lies in the forward, backward, and optimization steps of large neural networks. OrbDiff with SNIS, by contrast, computes the group-averaged target entirely outside the network, making it extremely efficient.
>
> In practice, the additional memory and computation required for sampling and averaging over group elements are negligible relative to the overall training cost. For instance, in our ProteinA experiments—one of the largest protein diffusion models—we use 10,000 SO(3) samples per training example. The added memory is just ~40MB per GPU, compared to ~54GB used during training.
>
> Importantly, we also measured the runtime overhead directly, as shown in Figure 5a. Even with 10,000 orbit samples, OrbDiff adds only 0.1% overhead to the total training time for ProteinA. To put this into perspective, in a 24-hour training run, this amounts to just 1.4 additional minutes—demonstrating that the method is highly efficient in practice.
>
> We will update the paper to highlight not only the minimal training-time overhead (reported in Figure 5a) but also the low memory footprint of OrbDiff, to better address concerns about its scalability.
>
> ### **Question #5: What if we do not bias?**
>
> Firstly, thanks to the reviewer's question, we take a closer look at how bias is OrbDiff and we find an interesting finding that **although OrbDiff's SNIS estimation of $\mathbb{E}[x_0 | x_t]$ is biased due to the choice of an efficient proposal distribution, the resulting gradient estimator is in fact unbiased**. This happens because the bias cancels out when computing the outer expectation of the gradient.
>
> We provide a detailed proof of this result in **our response to Reviewer 1 (j92t).**
>
> Given this, we believe that the performance difference between using a biased versus an unbiased proposal for estimating $\mathbb{E}[x_0 | x_t]$ is minimal when the sampling budget is sufficiently large (since both of them produce unbiased gradients). However, employing an unbiased proposal is both inefficient and not straightforward. For instance, computing the target for the loss function of a minibatch under an unbiased proposal requires sampling $x_0$ from outside the minibatch, complicating implementation. Moreover, as analyzed in the paper, the additional information gained from sampling external orbits is often negligible.
>
> Given that **OrbDiff yields an unbiased estimator of the gradient**, we believe it is still valuable to include an ablation comparing the performance of biased and unbiased proposals.
>
> We evaluated our approach on the Crystal Structure Prediction task using the Perov-5 dataset (DiffCSP Perov-5), which involves modeling a conditional distribution: given a list of atom types, generate a valid 3D crystal structure. Since an atom list can correspond to multiple valid structures, the task is inherently generative. In the Perov-5 training set, 3,330 out of 8,026 atom lists map to two distinct structures; the rest map to one.
>
> For an atom list with two valid structures, $x_0$ and $x_0^\prime$. Estimating $\mathbb{E}[x_0∣x_t]$ ideally requires sampling from both orbits. However, OrbDiff’s biased proposal samples only from $x_0$​'s orbit.
>
> We conduct an ablation study with the following model variants:
> - DiffCSP_bp_1k: Use biased proposal distribution with 1000 samples from the orbit of $x_0$.
> - DiffCSP_up_1k: Use unbiased proposal distribution with 1000 samples from both orbits of $x_0$ and $x_0^\prime$.
> - DiffCSP_up_2k: Use unbiased proposal distribution with 2000 samples from both orbits of $x_0$ and $x_0^\prime$.
>
> | Setting                          | Match Rate (↑)     | RMSD (↓)           |
> |----------------------------------|------------------|-----------------|
> | DiffCSP_bp_1k       |   **52.29**   |  0.078   |
> | DiffCSP_up_1k   |   52.02   |  0.090  |
> | DiffCSP_up_2k   |   52.13   |  **0.072**  |
>
> From the table, we observe that with the same sampling budget of 1,000, the model trained using the **biased** estimate of  $\mathbb{E}[x_0 \mid x_t]$ outperforms the one using the **unbiased** estimate. When doubling the sampling budget to 2,000, the unbiased variant (DiffCSP_up_2k) achieves a lower RMSD and a Match Rate close to the biased variant (DiffCSP_bp_1k), indicating that both approaches are effective depending on the target metric. This aligns with our theoretical analysis: both biased and unbiased estimates lead to the same **unbiased gradient in expectation**.

---

> ### Comment · Reviewer_LfDH · 2025-08-03
> **Response to authors**
>
> Dear authors, I would like to thank you for the detailed response to each of my questions. I have raised my rating to 5 -- this may not be reflected on your side immediately due to a possible change in the rule this year. I also recommend that the authors incorporate the responses to their revision to strengthen the contributions of the paper: for example, your response to Weakness 1 and Question 4, and the others.

---

> > ### Author Response · Authors · 2025-08-03
> > **Response to the Reviewer**
> >
> > Dear Reviewer,
> >
> > Thank you very much for your thoughtful feedback and for raising your score. We truly appreciate the time you took to engage deeply with our work and your constructive suggestions.
> >
> > We will make sure to incorporate the points discussed in our response—particularly regarding Weakness 1 and Question 4—into the revised version to strengthen the paper’s contributions.
> >
> > Thank you again for your support.

---

### Official Review · Reviewer_yVWb · 2025-07-01

**Clarity:** 3
**Significance:** 3
**Originality:** 2
**Rating:** 5
**Confidence:** 3

**Summary:**

This paper introduces Orbit Diffusion, a training framework for denoising diffusion models that respects symmetry in data, such as molecular or protein structures with SE(3) invariance. By interpreting data augmentation as a Monte Carlo approximation of a symmetrized loss and applying Rao–Blackwellization, the authors derive a low-variance gradient estimator that remains compatible with both equivariant and non-equivariant architectures. The method constructs denoising targets by averaging over symmetry group orbits using importance sampling, leading to improved stability and convergence.

**Questions:**

1. Can Orbit Diffusion handle approximate or partial symmetries—i.e., when the symmetry group is only approximately valid in the data?
2. How sensitive is Orbit Diffusion to the choice of group sampling distribution $\nu_t(g)$, especially in high-dimensional or non-compact groups like SE(3)?
3. How does the method handle symmetries that act only on a subset of the input (e.g., local symmetries, hierarchical groups)?
4. What would it take to extend this method to learn symmetry when the group is not known a priori (e.g., data-driven symmetry discovery)?

**Ethical Concerns:**

["NO or VERY MINOR ethics concerns only"]

**Final Justification:**

The authors have thoroughly addressed all of my concerns, and I have adjusted my rating accordingly.

**Limitations:**

Yes!

**Paper Formatting Concerns:**

None!

**Quality:**

3

**Strengths And Weaknesses:**

STRENGTHS
1. Introduces a provably lower-variance gradient estimator, grounded in classical statistical theory.
2. Provides a clear theoretical link between data augmentation and group symmetrization.
3. Enforces equivariance at the loss level, enabling training of both equivariant and non-equivariant models.
4. Proposes self-normalized importance sampling (SNIS) over group actions, which is tractable and sample-efficient.
5. Adds negligible computational cost to large models.

WEAKNESSES
1. The paper qualitatively discusses the bias–variance trade-off, but does not offer rigorous bounds or quantitative guarantees on the bias introduced. Figure 11 shows small empirical bias, but only for a toy molecule with 8 conformers. No real-world downstream impact of this bias is analyzed.
2. Sampling from large symmetry groups (e.g., permutations, SE(3), periodic translations) requires thousands of samples (e.g., 10,000 for SO(3) in PROTEINA), which may not be feasible in large-scale 3D scenes or protein assemblies. The paper provides no ablation on the trade-off between number of orbit samples and performance (e.g., how many samples are needed before diminishing returns). No hardware or memory scaling benchmarks are provided.
3. For several benchmarks (e.g., GEOM-QM9, MP-20), the evaluation is missing standard deviation/error bars, especially for high-variance metrics like AMR or RMSD. In some cases (e.g., ETFLOW), the reported baseline results could not be reproduced, raising concerns about reproducibility and benchmarking fairness.

---

> ### Author Rebuttal · Authors · 2025-07-29
>
> Thank you for your valuable feedback and for pointing out the areas of improvement in our manuscript.
>
> ### **Weakness #1: More insight and guarantee about the bias**
> Thank you for raising this point. Upon revisiting our analysis, we realized that although OrbDiff's SNIS estimation of $\mathbb{E}[x_0 \mid x_t]$ is biased, **the resulting gradient estimator is actually unbiased**. This happens because the bias cancels out when computing the outer expectation of the gradient. In the original paper, the term ``bias'' referred only to the approximation of $\mathbb{E}[x_0 \mid x_t]$, not to the gradient itself. This distinction is crucial, as training depends on the unbiasedness of the gradient.
>
> To further clarify, Figure~11 is meant to illustrate how well the OrbDiff target $\phi^*$ approximates $\mathbb{E}[x_0 \mid x_t]$ in a toy setting---not to assess bias in the gradient estimator. The quantity that truly matters for training is the gradient bias, which we now prove to be zero.
>
> Due to space limitations, we include the full proposition and proof in our **response to Reviewer 1 (j92t)**—please refer to that for details.
>
> ### **Weakness #2: Scalability and Efficiency of Symmetry Group Sampling**
> Thank you for the thoughtful question. We agree that scalability is an important concern for large symmetry groups. However, it's worth highlighting that the main cost in large-scale training typically lies in the forward, backward, and optimization steps of large neural networks. OrbDiff, by contrast, computes the group-averaged target entirely outside the network, making it extremely efficient.
>
> In practice, the additional memory and computation required for sampling and averaging over group elements are negligible relative to the overall training cost. For instance, in our ProteinA experiments—one of the largest protein diffusion models—we use 10,000 SO(3) samples per training example. The added memory is just ~40MB per GPU, compared to ~54GB used during training.
>
> Importantly, we also measured the runtime overhead directly, as shown in Figure 5a. Even with 10,000 orbit samples, OrbDiff adds only 0.1% overhead to the total training time for ProteinA. To put this into perspective, in a 24-hour training run, this amounts to just 1.4 additional minutes—demonstrating that the method is highly efficient in practice.
>
> We will update the paper to highlight not only the minimal training-time overhead but also the low memory footprint of OrbDiff, to better address concerns about its scalability.
>
> To further assess downstream performance, we conducted an ablation study on the Text-guided Crystal Structure Prediction task using the Perov-5 dataset (TGDMat Perov-5), where the goal is to generate a crystal structure given a set of atoms and a natural language description of a plausible material. In this study, we sampled uniformly from the periodic translation group (OrbDiff\_U) to demonstrate the effect of sampling size on performance.
>
> **Performance on the TGDMat Perov-5.**
> | # Samples | Match Rate (↑)    | RMSD (↓)        |
> |-----------|---------------|--------------|
> | 10        | 63.38  | 0.062 |
> | 100       | 64.28  | 0.055 |
> | 1000      | 65.57       | **0.054**         |
> | 5000      | **65.60**   | 0.056 |
>
> As shown above, increasing the number of samples from 10 to 1,000 improves the match rate from 63.38 to 65.57 and reduces RMSD from 0.062 to 0.054. However, beyond 1,000 samples, performance has converged, with no significant improvements observed at 5,000 samples. Therefore, using 1,000 samples offers a good balance between performance and efficiency.
>
> ### **Weakness #3: Reproducibility and Statistical Rigor in Benchmark Evaluation**
>
> Thank you for highlighting this important point. While prior work (including ETFlow and others) typically omits such statistics, we agree with the reviewer that including them helps better assess result variability and statistical significance.
>
> In response, during the rebuttal phase, we have managed to add the standard deviations for the QM9 benchmark. For each method, we perform evaluation three times and report the mean and standard deviation, shown in the following table:
>
> **Molecular conformer generation performance on GEOM-QM9.**
>
> \* Reported in the original paper.  † Obtained using the published checkpoint. ‡ We train the public implementation from scratch.
>
> | Model | COV\@0.1-R_mean (↑) | COV\@0.1-R_median (↑) | AMR-R_mean (↓) | AMR-R_median (↓) | COV\@0.1-P_mean (↑) | COV\@0.1-P_median (↑) | AMR-P_mean (↓)  | AMR-P_median (↓)|
> |-------|----------------|------------------|------------------|--------------------|------------------|--------------------|--------------------|----------------------|
> | GeoMol† | 28.4±0.9 | 0.0±0.0 | 0.224±0.001 | 0.192±0.002 | 20.8±0.2 | 0.0±0.0 | 0.270±0.001 | 0.240±0.004 |
> | Torsional Diff.† | 37.0±0.6 | 23.7±1.6 | 0.178±0.000 | 0.146±0.001 | 27.3±0.4 | 12.0±0.9 | 0.220±0.001 | 0.193±0.002 |
> | MCF† | 80.9±0.9 | **100.0±0.0** | 0.102±0.002 | 0.050±0.001 | 78.7±0.1 | **94.4±0.6** | **0.112±0.003** | 0.052±0.002 |
> | ETFlow* | - | - | 0.073 | 0.047 | - | - | 0.098 | 0.039 |
> | ETFlow† | 80.2±0.8 | **100.0±0.0** | 0.097±0.004 | 0.035±0.002 | 74.4±0.4 | 83.3±0.0 | 0.143±0.003 | 0.070±0.004 |
> | ETFlow‡ | 81.9±0.6 | **100.0±0.0** | 0.089±0.006 | 0.036±0.002 | 74.8±0.2 | 84.5±1.1 |0.143±0.002 | 0.064±0.002 |
> | + [OrbDiff] | **85.3±0.3** | **100.0±0.0** | **0.073±0.001** | **0.027±0.001** | **80.0±0.2** | 93.2±0.6 | **0.112±0.000** | **0.043±0.002** |
>
> ETFLow + [OrbDiff] consistently outperforms state of the art models on various metrics. Notably, we observe that the standard deviations for AMR, especially for OrbDiff, are relatively small, indicating the stability in evaluation. If the reviewer still feels strongly about including standard deviations, we are happy to accommodate that in the revised version of the paper.
>
> Regarding ETFLow - reproducibility:
> In the current version of the paper (Table 2), we have already taken care to address the reproducibility of ETFLOW by reporting three separate results:
> (i) the numbers reported in the original ETFlow paper,
> (ii) results obtained by running the official ETFlow code from scratch, and
> (iii) results obtained using a model checkpoint shared with us directly by the authors.
>
>
> ### **Question #1: Orbit Diffusion and approximate or partial symmetries.**
> OrbDiff can work with partial symmetries as long as there is an efficient method for sampling from the corresponding set of group actions. For instance, if we want to restrict equivariance up to some degree of rotation, or if we have only partial permutation equivariance. The former by simple restrictions of existing algorithms for sampling rotations, the latter with the product replacement algorithm.
>
> Extending the framework to explicitly model approximate or data-driven symmetries is a promising direction for future work. Such extensions could involve adaptive group selection or learning symmetry parameters jointly with the model, thereby relaxing the assumption of exact group structure. This is, however, beyond the scope of our work.
>
> ### **Question #2: The sensitivity of OrbDiff with the choice of group sampling distribution.**
>
> We thank the reviewer for the insightful question. As noted in lines 69--70 of the paper, we focus on locally compact isometry groups, which cover a broad range of practical applications. Extending the method to non-compact groups is certainly possible and presents an interesting direction for future work. In high-dimensional groups, it may be necessary to increase the number of group samples to ensure sufficient coverage of the group distribution.
>
> Nevertheless, we note that standard diffusion training can be viewed as sampling only the identity element of the group. In contrast, OrbDiff samples multiple group elements, which we have shown reduces the variance of the gradient estimator without introducing bias. Given this, along with the efficiency of OrbDiff, we believe it is generally advisable to prefer OrbDiff over standard training whenever group symmetries are known and we can sample from the set of group actions.
>
>
> ### **Question #3: Orbit Diffusion and local symmetries or hierarchical groups.**
>
> We thank the reviewer for this interesting question. In principle, the framework could be extended to handle local symmetries by defining group actions on different regions and applying our SNIS-based estimator accordingly. Hierarchical symmetries could be approached via multi-scale or nested sampling strategies. We consider these valuable directions for future work.
>
> ### **Question #4: Extending this method to learn symmetry when the group is not known priori.**
>
> We thank the reviewer for raising the possibility of learning symmetries from data. While our method assumes known group actions, it is orthogonal and potentially complementary to symmetry discovery methods such as [1], [2] or [3]. These methods offer promising directions for automatic symmetry identification. Future work could explore coupling our loss-level equivariant framework with a learned group structure, possibly enabling generalization to approximate or local symmetries.
>
> [1] Generative adversarial symmetry discovery
>
> [2] AtlasD: Automatic Local Symmetry Discovery
>
> [3] Automatic symmetry discovery with lie algebra convolutional network

---

> > ### Comment · Reviewer_yVWb · 2025-08-05
> >
> > Thank you for addressing my concerns. I have revised my rating accordingly.

---

### Official Review · Reviewer_j92t · 2025-07-04

**Clarity:** 3
**Significance:** 4
**Originality:** 4
**Rating:** 5
**Confidence:** 4

**Summary:**

The paper proposes Orbit Diffusion as a Rao-Blackwellized gradient estimator to train equivariant diffusion models. The core idea is to Rao-Blackwellize the targets, not the denoiser, by conditioning on the symmetrized data distribution.

For a practical computation of the conditional expectation, self-normalised importance sampling is used with a user-defined proposal.

**Questions:**

How sensitive are experiment results to the choice of biased or unbiased proposals?

**Ethical Concerns:**

["NO or VERY MINOR ethics concerns only"]

**Limitations:**

See weaknesses

**Paper Formatting Concerns:**

None.

**Quality:**

4

**Strengths And Weaknesses:**

Strengths

The paper is written and presented well, with sufficient background on equivariance, group symmetrization and diffusion models to describe the proposed method.

The derivation of the Rao-Blackwellized gradient estimator is detailed.

Empirical results support the proposed method.

The approximation strategy for the conditional expectation of $x_0 \sim \hat{q}_t^{G}(x_0|x_t)$ is stated clearly as the main limitation and an avenue for future work.

Weaknesses

While the use of a biased proposal is well-motivated (in lines 170-171), experiment results should include an ablation study for using a biased proposal vs an unbiased proposal.

---

> ### Author Rebuttal · Authors · 2025-07-29
>
> Thank you for your valuable feedback and for pointing out the areas of improvement in our manuscript.
>
> ### **Weakness & Question #1: Ablation study of biased proposal and unbiased proposal**
>
> Thank you for raising this insightful point. We agree that an ablation study would strengthen the paper by illustrating how the biased proposal distribution impacts final performance. To clarify, whenever we refer to the proposal or proposal distribution, we mean the proposal used in the self-normalized importance sampler (SNIS) for estimating the inner expectation $\mathbb{E}[x_0 \mid x_t]$. This estimate in OrbDiff is biased because the proposal’s support does not fully cover that of the target distribution $\hat{q}_t^G(x_0 \mid x_t)$. However, as we will show later, this bias cancels out when computing the gradient, making OrbDiff an **unbiased estimator of the training gradient.**
>
> We evaluated our approach on the Crystal Structure Prediction task using the Perov-5 dataset (DiffCSP Perov-5), which involves modeling a conditional distribution: given a list of atom types, generate a valid 3D crystal structure. Since an atom list can correspond to multiple valid structures, the task is inherently generative. In the Perov-5 training set, 3,330 out of 8,026 atom lists map to two distinct structures; the rest map to one.
>
> For an atom list with two valid structures, $x_0$ and $x_0^\prime$. Estimating $\mathbb{E}[x_0∣x_t]$ ideally requires sampling from both orbits. However, OrbDiff’s biased proposal samples only from $x_0$​'s orbit.
>
> We conduct an ablation study with the following model variants:
> - DiffCSP_bp_1k: Use biased proposal distribution with 1000 samples from the orbit of $x_0$.
> - DiffCSP_up_1k: Use unbiased proposal distribution with 1000 samples from both orbits of $x_0$ and $x_0^\prime$.
> - DiffCSP_up_2k: Use unbiased proposal distribution with 2000 samples from both orbits of $x_0$ and $x_0^\prime$.
>
> | Setting                          | Match Rate (↑)     | RMSD (↓)           |
> |----------------------------------|------------------|-----------------|
> | DiffCSP_bp_1k       |   **52.29**   |  0.078   |
> | DiffCSP_up_1k   |   52.02   |  0.090  |
> | DiffCSP_up_2k   |   52.13   |  **0.072**  |
>
> From the table, we observe that with the same sampling budget of 1,000, the model trained using the **biased** estimate of  $\mathbb{E}[x_0 \mid x_t]$ outperforms the one using the **unbiased** estimate. When doubling the sampling budget to 2,000, the unbiased variant (DiffCSP_up_2k) achieves a lower RMSD and a Match Rate close to the biased variant (DiffCSP_bp_1k), indicating that both approaches are effective depending on the target metric. This aligns with our theoretical analysis: both biased and unbiased estimates lead to the same **unbiased gradient in expectation** (see the proof below).
>
>
> ## **Additional Clarification [Relevant to all Reviewers]:**
>
> In revisiting our analysis during the rebuttal—prompted particularly by Reviewer 2’s question on the potential bias of OrbDiff—we discovered an important theoretical property of our method that was not recognized during the initial writing of the paper: the **OrbDiff gradient estimator is unbiased**, even though the OrbDiff SNIS estimation of the inner expectation $\mathbb{E}[x_0 | x_t]$ is biased.
>
> Since OrbDiff samples $x_0$ only from the orbit that gave rise to $x_t$, rather than from the full symmetrized distribution $\hat{q}^G(x_0)$, the support of OrbDiff's proposal distribution does not cover that of the original distribution. As a result, this approximation introduces bias in estimating the target value $\mathbb{E}[x_0 \mid x_t]$. **Crucially, however, the resulting gradient estimator remains unbiased.**
>
> This distinction is important: for training to be statistically sound, what ultimately matters is the unbiasedness of the gradient.
>
> To summarize, and to have a better clarity, we have the following table:
> #### Distinguishing RB Estimator vs. OrbDiff
> | Aspect       | Rao-Blackwell Estimator | OrbDiff |
> |--------------|--------------------------|---------|
> | **Target** | Exact $\mathbb{E}_{x_0 \sim \hat{q}_t^G(x_0 \mid x_t)}[x_0]$ | Monte Carlo estimate using samples **only from the orbit** of the true $x_0$ |
> | **Sampling** | From full $\hat{q}^G(x_0)$, includes all possible orbits | From the orbit of the training sample $x_0$ |
> | **Estimator** | Unbiased for both target and gradients | Biased estimator of the target but **unbiased gradients** |
> | **Practicality** | Inefficient and nontrivial to implement | Efficient and easy to implement |
>
> Our goal is to estimate the following gradient (Eq.(6) in the paper):
>
> $$
> \nabla_\phi \mathcal{L}_t^{G}(\phi) = 2\mathbb{E} _{x_t \sim \hat{q}_t^G(x_t)} \left[  \phi(x_t, t) - \mathbb{E} _{x_0 \sim \hat{q}_t^G(x_0 \mid x_t)} [x_0]  \right].
> $$
>
> Our proposed Rao-Blackwell loss function (Eq.(13) in the paper) has the same gradient as $\mathcal{L} _t^G(\phi)$, but with reduced variance due to the use of the expectation $\mathbb{E} _{x_0 \sim \hat{q}_t^G(x_0 \mid x_t)}[x_0]$ :
>
> $$
> \mathcal{L}_t^{\mathrm{RB}}(\phi) = \mathbb{E} _{x_0' \sim \hat{q}^G(x'_0)} \mathbb{E} _{x_t \sim \hat{q} _t^G(x_t \mid x_0')} \left[\lVert \phi(x_t, t) - \mathbb{E} _{x_0 \sim \hat{q}_t^G(x_0 \mid x_t)}[x_0]   \lVert^2 \right].
> $$
>
> However, computing $\mathbb{E}_{x_0 \sim \hat{q}_t^G(x_0 \mid x_t)} [x_0]$ can be inefficient and nontrivial—for example, it may require sampling from data orbits not present in the current batch. To address this, we introduce OrbDiff, which uses a biased proposal distribution to approximate this expectation using only samples from the orbit of the $x_0$ that generated $x_t$. This yields the alternative target:
>
> $$
> \phi^*(x_0, x_t, t) = \frac{1}{Z(x_0, x_t)} \int_{G} (g \circ x_0) \hat{q}_t^G(g \circ x_0 \mid x_t) \mathrm{d}\mu_G(g),
> $$
>
> where the normalization term is
>
> $$
> Z(x_0, x_t) = \int_{G} \hat{q}_t^G(g \circ x_0 \mid x_t) \mathrm{d}\mu_G(g).
> $$
>
> This matches Eq.(14) in the paper, which can be verified via straightforward transformations.
>
> Although $\phi^*$ differs from
> $\mathbb{E} _{x _0 \sim \hat{q} _t^G(x _0 \mid x _t)} [x _0]$,
>
> we show that replacing the Rao-Blackwell target with $\phi^*$ yields a loss whose gradient still matches the original gradient. This relies on the assumption that the forward conditional distribution is equivariant under the group action (such as Gaussian kernel), i.e.,
>
> $$
> \hat{q}_t^G(g \circ x_t \mid g \circ x_0) = \hat{q}_t^G(x_t \mid x_0), \quad \forall g \in G,
> $$
>
> Under this assumption, the OrbDiff loss is given by:
>
> $$
> \mathcal{L}_t^{\mathrm{OrbDiff}}(\phi) = \mathbb{E} _{x_0 \sim \hat{q}^G(x_0)} \mathbb{E} _{x_t \sim \hat{q}_t^G(x_t \mid x_0)} \left[ \lVert \phi(x_t, t) - \phi^*(x_0, x_t, t) \lVert^2 \right] \\
> $$
>
> $$
> \qquad \qquad = \mathbb{E} _{x_t \sim \hat{q}_t^G(x_t)} \mathbb{E} _{x_0 \sim \hat{q}_t^G(x_0 \mid x_t)} \left[ \lVert \phi(x_t, t) - \phi^*(x_0, x_t, t) \lVert^2 \right].
> $$
>
> Taking the gradient with respect to $\phi$, we obtain:
>
> $$
> \nabla_\phi \mathcal{L}_t^{\mathrm{OrbDiff}}(\phi) = 2 \mathbb{E} _{x_t \sim \hat{q}_t^G(x_t)} \mathbb{E} _{x_0 \sim \hat{q}_t^G(x_0 \mid x_t)} \left[ \phi(x_t, t) - \phi^*(x_0, x_t, t) \right]
> $$
>
> $$
> \qquad \qquad = 2 \mathbb{E} _{x_t \sim \hat{q}_t^G(x_t)} \left[ \phi(x_t, t) - \mathbb{E} _{x_0 \sim \hat{q}_t^G(x_0 \mid x_t)} [\phi^*(x_0, x_t, t)] \right].
> $$
>
> Thus, to ensure that OrbDiff yields the correct gradient, we will show $\nabla _\phi \mathcal{L} _t^{G}(\phi) = \nabla _\phi \mathcal{L} _t^{\mathrm{OrbDiff}}(\phi)$. It is suffice to verify:
>
> $$
> \mathbb{E} _{x_0 \sim \hat{q}_t^G(x_0 \mid x_t)} [x_0] = \mathbb{E} _{x_0 \sim \hat{q}_t^G(x_0 \mid x_t)} [\phi^*(x_0, x_t, t)].
> $$
>
> We compute:
>
> $$
> RHS = \int _\Omega \int _G \phi ^*(g' \circ x _0, x _t, t) \hat{q} _t^G(g' \circ x _0 \mid x_t)  \mathrm{d}\mu _G(g')  \mathrm{d}x_0.
> $$
>
> Substituting the expression for $\phi^*$ and applying the change of variables $g \mapsto g \cdot g'$ with left-invariant Haar measure $\mu_G$, and since $Z(x_0, x_t)$ is invariant to group transformations of $x_0$, we have:
>
> $$
> RHS = \int_\Omega \int_G \left[ \frac{1}{Z(g' \circ x_0, x_t)} \int_G (g \circ [g' \circ x_0]) \hat{q}_t^G(g \circ [g' \circ x_0] \mid x_t) \mathrm{d}\mu_G(g) \right] \hat{q}_t^G(g' \circ x_0 \mid x_t) \mathrm{d}\mu_G(g') \mathrm{d}x_0
> $$
>
> $$
> = \int_\Omega \int_G \left[ \frac{1}{Z(x_0, x_t)} \int_G (g \circ x_0) \hat{q}_t^G(g \circ x_0 \mid x_t) \mathrm{d}\mu_G(g) \right] \hat{q}_t^G(g' \circ x_0 \mid x_t) \mathrm{d}\mu_G(g') \mathrm{d}x_0
> $$
>
> $$
> = \int_\Omega \left[ \frac{1}{Z(x_0, x_t)} \int_G (g \circ x_0) \hat{q}_t^G(g \circ x_0 \mid x_t) \mathrm{d}\mu_G(g) \right] \left[ \int_G \hat{q}_t^G(g' \circ x_0 \mid x_t) \mathrm{d}\mu_G(g') \right] \mathrm{d}x_0
> $$
>
> $$
> = \int_\Omega \left[ \frac{1}{Z(x_0, x_t)} \int_G (g \circ x_0) \hat{q}_t^G(g \circ x_0 \mid x_t) \mathrm{d}\mu_G(g) \right] Z(x_0, x_t) \\mathrm{d}x_0
> $$
>
> $$
> = \int_\Omega \int_G (g \circ x_0) \hat{q}_t^G(g \circ x_0 \mid x_t) \mathrm{d}\mu_G(g) \mathrm{d}x_0
> $$
>
> $$
> = \mathbb{E}_{x_0 \sim \hat{q}_t^G(x_0 \mid x_t)} [x_0].
> $$
>
> Therefore, despite $\phi^*$ not being equal to $\mathbb{E}[x_0 | x_t]$, the gradient induced by the OrbDiff loss matches the desired gradient. **OrbDiff thus provides an unbiased estimate of $\nabla_\phi \mathcal{L}_t^{G}(\phi)$**, while using only samples from the orbit of $x_0$.

---

> > ### Author Response · Authors · 2025-08-05
> >
> > Thank you for your thorough and insightful review. As the rebuttal period is coming to its end, we are reaching out to see if you have any additional questions or concerns that we can address. Thank you for your valuable feedback, and we sincerely anticipate your response.

---

### Note · Authors · 2025-08-12

We thank the reviewers for their thorough evaluation and valuable feedback. The reviewers consistently acknowledged the following strengths:

- A rigorous theoretical foundation based on Rao–Blackwellization with provable variance reduction.
- Broad applicability to both equivariant and non-equivariant models.
- Strong empirical validation across multiple benchmarks.
- Conceptual simplicity and minimal computational overhead, ensuring practical scalability.

The primary concerns raised and our responses are as follows:

- **Potential bias of OrbDiff:** We performed a rigorous theoretical analysis showing that, although the inner conditional expectation estimate is biased, the overall gradient estimator remains unbiased.
- **Computational and memory overhead:** We clarified that OrbDiff’s overhead is negligible in practice, even for large-scale models.
- **Presentation and motivation:** We appreciate the suggestions to improve presentation clarity and to better motivate the focus on variance reduction, which we intend to incorporate in the revised manuscript.

We sincerely thank the reviewers for their insightful and constructive input, which has greatly improved the quality of our work.

---

### Decision · Program_Chairs · 2025-09-17

**Decision:**

Accept (poster)

**Comment:**

All the four reviewers lean towards acceptance. That said, reviewers have raised several areas for improvement that the authors should address carefully in the final version, including adding clarifications, new analysis and enhancing the presentation and organization of the paper to minimize confusion. My recommendation is acceptance.